# Accountability in Offline Reinforcement Learning: Explaining Decisions with a Corpus of Examples

**Hao Sun,** * **Alihan Hüyük, Daniel Jarrett, Mihaela van der Schaar**
Department of Applied Mathematics and Theoretical Physics
University of Cambridge

## Abstract

Learning controllers with offline data in decision-making systems is an essential area of research due to its potential to reduce the risk of applications in real-world systems. However, in responsibility-sensitive settings such as healthcare, decision accountability is of paramount importance, yet has not been adequately addressed by the literature. This paper introduces the Accountable Offline Controller (AOC) that employs the offline dataset as the Decision Corpus and performs accountable control based on a tailored selection of examples, referred to as the Corpus Subset. AOC operates effectively in low-data scenarios, can be extended to the strictly offline imitation setting, and displays qualities of both conservation and adaptability. We assess AOC's performance in both simulated and real-world healthcare scenarios, emphasizing its capability to manage offline control tasks with high levels of performance while maintaining accountability.

## 1 Introduction

In recent years, offline control that uses pre-collected data to generate control policies has gained attention due to its potential to reduce the costs and risks associated with applying control algorithms in real-world systems [1], which are especially advantageous in situations where real-time feedback is challenging or expensive to obtain [2–5]. However, existing literature has primarily focused on enhancing learning performance, leaving the need for accountable and reliable control policies in offline settings largely unaddressed, particularly for high-stakes, responsibility-sensitive applications.

However, in many critical real-world applications such as healthcare, the challenge is beyond enhancing policy performance. It requires the decisions made by learned policies to be transparent, traceable, and justifiable. Yet those essential properties, summarized as Accountability, are left largely unaddressed by existing literature.

In our context, we use *Accountability* to indicate the existence of *a supportive basis for decision-making*. For instance, in tumor treatment using high-risk options like radiotherapy and chemotherapy, the treatment decisions should be based on the successful outcomes experienced by previous patients who share similar conditions and were given the same medication. Another concrete illustrative example is the allocation decisions of ventilator machines. The decision to allocate a ventilator should be accountable, in the way that it juxtaposes the potential consequences of both utilization and non-utilization and provides a reasonable decision on those bases. In those examples, the ability to refer to existing cases that support current decisions can enhance reliability and facilitate reasoning or debugging of the policy. To advance offline control towards real-world responsibility-sensitive applications, five properties are desirable: **(P1)** controllable **conservation**: this ensures the policy learning performance by avoiding aggressive extrapolation. **(P2) accountability**: as underscored by our prior examples, there is a need for a clear basis upon which decisions are made. **(P3)** suitability

---

* hs789@cam.ac.uk

37th Conference on Neural Information Processing Systems (NeurIPS 2023).

for **low-data** regimes: given the frequent scarcity of high-stake decision data, it's essential to have methods that perform well with limited data. **(P4) adaptability** to user specification: this ensures the policy can adjust to changes, like evolving clinical guidelines, allowing for tailored solutions. **(P5) flexibility** in Strictly Offline Imitation: this property ensures a broader applicability across various scenarios and data availability.

To embody all these properties, we need to venture beyond the current scope of literature focused on conservative offline learning. In our work:

1. Methodologically, we introduce the formal definitions and necessary concepts in accountable decision-making. We propose the Accountable Offline Controller (AOC), which makes decisions according to a decomposition on the basis of the representative existing decision examples.
2. Theoretically, we prove the existence and uniqueness of the decomposition under mild conditions, guiding the design of our algorithms.
3. Practically, we introduce an efficient algorithm that takes all the aforementioned desired properties into consideration and circumvented the computational difficulty.
4. Empirically, we verify and highlight the desired properties of AOC on a variety of offline control tasks, including five simulated continuous control tasks and one real-world healthcare dataset.

## 2 Preliminaries

**POMDP**   We develop our work under the general Partially Observable Markov Decision Process (POMDP) setting, denoted as a tuple $(\mathcal{X}, \omega, \mathcal{O}, \mathcal{A}, \mathcal{T}, \mathcal{R}, \gamma, \rho_0)$, where $\mathcal{X} \subseteq \mathbb{R}^{d_x}$ denotes the underlying $d_x$ dimensional state space, $\omega : \mathcal{X} \mapsto \mathcal{O}$ is the emission function that maps the underlying state into a $d_o$ dimensional observation $\mathcal{O} \subseteq \mathbb{R}^{d_o}$; $\mathcal{A} \subseteq \mathbb{R}^{d_a}$ is a $d_a$ dimensional action space, transition dynamics $\mathcal{T} : \mathcal{X} \times \mathcal{A} \mapsto \mathcal{X}$ controls the underlying transition between states given actions; and the reward function $\mathcal{R} : \mathcal{X} \mapsto \mathbb{R}$ maps state into a reward scalar; we use $\gamma$ to denote the discount factor and $\rho_0$ the initial state distribution. Note that the MDP is the case when $\omega$ is an identical mapping.

**Offline Control**   Specifically, our work considers the offline learning problem: a logged dataset $\mathcal{D} = \{o_t^i, a_t^i, r_t^i, o_{t+1}^i\}_{t=1,\dots,T}^{i=1,\dots,N}$ containing $N$ trajectories with length $T$ is collected by rolling out some behavior policies in the POMDP, where $x_0^i \sim \rho_0$ sampled from the initial state distribution and $o_0^i = \omega(x_0^i)$, $a_t^i \sim \pi_b^i$ sampled from unknown behavior policies $\pi_b^i$. We denote the observational transition history until time $t$ as $h_t$, such that $h_t = (o_{<t}, a_{<t}, r_{<t}) \in \mathcal{H} \subseteq \mathbb{R}^{(d_o+d_a+1)\cdot(t-1)}$.

**Learning Objective**   A policy $\pi : \mathcal{O} \times \mathcal{H} \mapsto \mathcal{A}$ is a function of observational transition history. The learning objective is to find $\pi$ that maximizes the expected cumulative return in the POMDP:

$$\max_\pi \mathbb{E}_{\tau \sim \rho_0, \pi, \mathcal{T}} \left[ \gamma^t r_t \right] \tag{1}$$

where $\tau \sim \rho_0, \pi, \mathcal{T}$ means the trajectories $\tau$ is generated by sampling initial state $x_0$ from $\rho_0$, and action $a$ from $\pi$, such that: $\tau = \{x_0, o_0 = \omega(x_0), a_0 = \pi(o_0, h_0 = \phi), x_1 = \mathcal{T}(x_0, a_0), r_0 = \mathcal{R}(x_1), ...\}$. A learned belief variable $b_t \in \mathbb{R}^{d_b}$ can be introduced to represent the underlying state $x_t$.

## 3 Accountable Control with Decision Corpus

### 3.1 Method Sketh: A Roadmap

The high-level core idea of our work is to introduce an example-based accountable framework for offline decision-making, such that the decision basis can be clear and transparent.

To achieve this, a naive approach would be to leverage the insights of Nearest Neighbors: for each action, this involves finding the most similar transitions in the offline dataset, and estimating the corresponding outcomes. Nonetheless, a pivotal hurdle arises in defining **similarity**, particularly when taking into account the intricate nature of trajectories, given both the observation space heterogeneity and the inherent temporal structure in decision-making. Compounding such a challenge, another difficulty arises in identifying the most **representative** examples and integrating the pivotal principle

of **conservation**, which is widely acknowledged to be essential for the success of offline policy learning.

Our proposed method seeks to address those challenges. We start by introducing the basic definitions to support a formal discussion of accountability: in Definition 3.1, we introduce the concept of Decision Corpus.

To address the similarity challenge, we showcase that a nice linear property (Property 3.2) generally exists (Remarks 3.3 & 3.4) when working in the belief space (Definition 3.5). This subsequently leads to a theoretical bound for estimation error (Proposition 3.8);

To address the representative challenge while obeying the principle of conservation, we underscore those examples that span the convex hull (Definition 3.6) and introduce the related optimization objective (Definition 3.7 & 3.9). In a nutshell, the intuition is to use a minimal set of representative training examples to encapsulate test-time decisions. Under mild conditions, we show the solution would exist and be unique (Proposition 3.10).

### 3.2 Understanding Decision-Making with a Subset of Offline Data

In order to perform accountable decision-making, we construct our approach upon the foundation of the example-based explanation framework [6, 7], which motivates us to define the offline decision dataset $\mathcal{D}$ as the *Decision Corpus*, and introduce the concept of *Corpus Subset* that is composed of a representative subset of the decision corpus. These corpus subsets will be utilized for comprehending control decisions. Formally:

**Definition 3.1** (Corpus Subset). A *Corpus Subset* $\mathcal{C}$ is defined as a subset of the decision corpus $\mathcal{D}$, indexed by $[C] := \{1, 2, ..., C\}$ — the natural numbers between 1 and $C$.

$$\mathcal{C} = \left\{ (o^c, a^c, h^c, v^c) \in \mathcal{D} \,\middle|\, c \in [C] \right\}. \tag{2}$$

In the equation above, the transition tuple $(o^c, a^c, h^c, v^c)$ can be obtained from the decision corpus. We absorb both superscripts ($i$) and subscripts ($t$) to the index ($c$) for the conciseness of notions. Later in our work, we use subscripts $t$ to denote the control time step. The variable $v^c \in \mathcal{V} \subseteq \mathbb{R}$ represents a performance metric that is defined based on the user-specified decision objective. e.g., $v^c$ may correspond to an ongoing cumulative return, immediate reward, or a risk-sensitive measure.

Our objective is to understand the decision-making process for control time rollouts with the decision corpus. To do this, a naive approach is to reconstruct the control time examples with the corpus subset $(o_t, a_t, h_t) = \sum_{c=1}^{C} w^c(o^c, a^c, h^c)$, where $w^c \in [0, 1]$ and $\sum_{c=1}^{C} w^c = 1$. Then the corresponding performance measures $v^c$ can be used to estimate $v_t | a_t$ — the control time return of a given decision $a_t$. Nonetheless, there are critical issues associated with this straightforward method:

1. Defining a distance metric for the joint observation-action-history space presents a challenge. This is not only due to the fact that various historical observations are defined over different spaces but also because the Euclidean distance fails to capture the structural information effectively.
2. The conversion from the observation-action-history space to the performance measure does not necessarily correspond to a linear mapping: $v_t \neq \sum_{c=1}^{C} w^c v^c$.

To address those difficulties, we propose an alternative approach called Accountable Offline Controller (AOC) that executes traceable offline control leveraging certain examples as decision corpus.

### 3.3 Linear Belief Learning for Accountable Offline Control

To alleviate the above problems, we first introduce a specific type of belief function $\boldsymbol{b} : \mathcal{O} \times \mathcal{A} \times \mathcal{H} \mapsto \mathcal{B} \subseteq \mathbb{R}^{d_b}$, which maps a joint observation-action-history input onto a $d_b$-dimensional belief space.

**Property 3.2** (Belief Space Linearity). The performance measure function $\boldsymbol{V} : \mathcal{O} \times \mathcal{A} \times \mathcal{H} \mapsto \mathcal{V}$ can always be decomposed as $\boldsymbol{V} = \boldsymbol{l} \circ \boldsymbol{b}$, where $\boldsymbol{b} : \mathcal{O} \times \mathcal{A} \times \mathcal{H} \mapsto \mathcal{B} \subseteq \mathbb{R}^{d_b}$ maps the joint observation-action-history space to a $d_b$ dimensional belief variable $b_t = \boldsymbol{b}(o_t, a_t, h_t)$ and $\boldsymbol{l} : \mathcal{B} \mapsto \mathcal{V}$ is a linear function that maps the belief $b_t$ to an output $\boldsymbol{l}(b_t) \in \mathcal{V}$.

*Remark* 3.3. This property is often the case with prevailing neural network approximators, where the belief state is the last activated layer before the final linear output layer.

*Remark* 3.4. The belief mapping function maps a $d_o + d_a + (d_o + d_a + 1) \cdot (t - 1)$ dimensional varied-length input into a fixed-length $d_b$ dimensional output. Usually, we have $1 = d_v < d_b \ll d_o + d_a + (d_o + d_a + 1) \cdot (t - 1)$. It is important to highlight that calculating the distance in the belief space is feasible, whereas it is intractable in the joint original space.

Based on Property 3.2, the linear relationship between the belief variable and the performance measure makes belief space a better choice for example-based decision understanding — this is because linearly decomposing the belief space also breaks down the performance measure, hence can guide action selection towards optimizing the performance metric. To be specific, we have

$$v(a_t|o_t, h_t) = \boldsymbol{l} \circ \boldsymbol{b}(o_t, a_t, h_t) = \boldsymbol{l} \circ \sum_{c=1}^{C} w^c \cdot \boldsymbol{b}(o^c, a^c, h^c) = \sum_{c=1}^{C} w^c \cdot \boldsymbol{l} \circ \boldsymbol{b}(o^c, a^c, h^c) = \sum_{c=1}^{C} w^c \cdot v^c \quad (3)$$

Although we have formally defined of the *Corpus Subset*, its composition has not been explicitly specified. We will now present the fundamental principles and techniques employed to create a practical *Corpus Subset*.

### 3.4 Selection of Corpus Subset

As suggested by Equation (3), if a belief state can be expressed as a linear combination of examples from the offline dataset, the same weights applied to the linear decomposition of the belief space can be utilized to decompose and synthesize the outcome. More formally, we define the convex hull spanned by the image of the Corpus Subset under the mapping function $\boldsymbol{b}$:

**Definition 3.5** (Belief Corpus Subset). A *Belief Corpus Subset* $\boldsymbol{b}(\mathcal{C})$ is defined by applying the belief function $\boldsymbol{b}$ to the corpus subset $\mathcal{C}$, with the corresponding set of values denoted as $v(\mathcal{C})$:

$$\boldsymbol{b}(\mathcal{C}), v(\mathcal{C}) = \left\{ b^c = \boldsymbol{b}(o^c, a^c, h^c), v^c \middle| (o^c, a^c, h^c, v^c) \in \mathcal{C} \right\} \subset \mathcal{B} \times \mathcal{V} \quad (4)$$

**Definition 3.6** (Belief Corpus Convex Hull). The *Belief Corpus Convex Hull* spanned by a corpus subset $\mathcal{C}$ with the belief corpus $\boldsymbol{b}(\mathcal{C})$ is the convex set

$$\mathcal{CB}(\mathcal{C}) = \left\{ \sum_{c=1}^{C} w^c b^c \middle| w^c \in [0, 1], b^c \in \boldsymbol{b}(\mathcal{C}), \forall c \in [C], \sum_{c=1}^{C} w^c = 1 \right\} \quad (5)$$

Note that an accurate belief corpus subset decomposition for a specific control time belief value $b_t$ may be unattainable when $b_t \notin \mathcal{CB}(\mathcal{C})$. In such situations, the optimal course of action is to identify the element $\tilde{b}_t$ within $\mathcal{CB}(\mathcal{C})$ that provides the closest approximation to $b_t$. Denoting the norm in the belief space as $|| \cdot ||_\mathcal{B}$, this equates to minimizing a corpus residual that is defined as the distance from the control time belief to the nearest point within the belief corpus convex hull:

**Definition 3.7** (Belief Corpus Residual). The *Belief Corpus Residual* given a control time belief variable $b_t$ and a belief corpus subset $\mathcal{C}$ is given as

$$r_\mathcal{C}(b_t) = \min_{\tilde{b}_t \in \mathcal{CB}(\mathcal{C})} ||b_t - \tilde{b}_t||_\mathcal{B} \quad (6)$$

In summary, when $b_t$ resides within $\mathcal{CB}(\mathcal{C})$, it can be decomposed using the belief corpus subset according to $b_t = \sum_{c=1}^{C} w^c b^c$. Here, the weights can be understood as representing the similarity and importance in reconstructing the control time belief variable through the belief corpus subset; otherwise, the optimizer of Equation (6) can be employed as a proxy for $b_t$. Under such circumstances, the **belief corpus residual acts as a gauge for the degree of extrapolation**. Given those definitions, we use the following proposition to inform the choice of an appropriate corpus subset.

**Proposition 3.8** (Estimation Error Bound for $v(a_t)$, *cf.* Crabbé et al. [7]). *Consider the belief variable* $b_t = \boldsymbol{b}(o_t, a_t, h_t)$ *and* $\hat{b}$ *the optimizer of Equation (6), the estimated value residual between* $\boldsymbol{l}(b_t)$ *and* $\boldsymbol{l}(\hat{b})$ *is controlled by the corpus residual:*

$$||\boldsymbol{l}(\hat{b}) - \boldsymbol{l}(b_t)||_\mathcal{V} \leq ||\boldsymbol{l}||_{\mathrm{op}} \cdot r_\mathcal{C}(b_t) \quad (7)$$

*where* $|| \cdot ||_\mathcal{V}$ *is a norm on* $\mathcal{V}$ *and* $||\boldsymbol{l}||_{\mathrm{op}} = \inf \{\lambda \in \mathbb{R}^+ : ||\boldsymbol{l}(b)||_\mathcal{V} \leq \lambda ||b||_\mathcal{B}\}$ *is the operator norm for the linear mapping.*

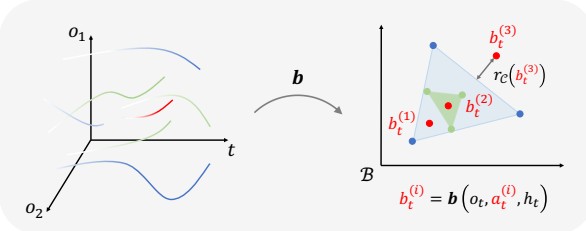

Figure 1: We illustrate the concepts of Corpus Residual and Minimal Hull in a 2-dim example. The belief function $\boldsymbol{b}$ maps trajectories in the offline dataset (colors of **blue** and **green** denote that they may come from different behavior policies) onto the belief space. The **control trajectory** with different sampled actions at time $t$ can also be mapped to the belief space. Some of those sampled action candidates (e.g., $a_t^{(1)}, a_t^{(2)}$) can be well-supported by a corpus subset, while in another case ($a_t^{(3)}$) the corpus residual manifests the extrapolation error. Among the three plotted beliefs generated by candidate actions, $b_t^{(3)}$ is out of the convex hulls that can be spanned by any decision corpus, $b_t^{(2)}$ has the minimal corpus (green shaded area), $b_t^{(1)}, b_t^{(2)}$ are also in a larger corpus hull (blue shaded area) but it is not ideal.

Proposition 3.8 indicates that minimizing the belief corpus residual also minimizes the estimation error. In order to approximate the belief variables in control time accurately, we propose to use the corpus subset that spans the minimum convex hull in the belief space.

**Definition 3.9** (Minimal Hull and Minimal Corpus Subset). Denoting the corpus subsets whose belief convex hull contains a belief variable $b_t$ by

$$\mathcal{C}(b_t) = \left\{ \mathcal{C} \subseteq \mathcal{D} \middle| b_t \in \mathcal{CB}(\mathcal{C}) \right\}$$

Among those subsets, the one that contains *maximally* $d_b + 1$ examples and has the smallest hyper-volume forms the *minimal hull* is called the *minimal corpus subset* (w.r.t. $b_t$), denoted by $\tilde{\mathcal{C}}(b_t)$:

$$\tilde{\mathcal{C}}(b_t) := \arg\min_{\mathcal{C}} \int_{\mathcal{CB}(\mathcal{C})} dV$$

**Proposition 3.10** (Existence and Uniqueness). *Consider the belief variable $b_t = \boldsymbol{b}(o_t, a_t, h_t)$, if $r_{\mathcal{C}}(b_t) \leq 0$ holds for $\mathcal{C} = \mathcal{D}$, then $\tilde{\mathcal{C}}(b_t)$ exists and the decomposition on the corresponding minimal convex hull exists and is unique.*

In this work, we use the corpus subset that constructs the minimal (belief corpus convex) hull to represent the control time belief variable $b_t$. Those concepts are illustrated in Figure 1.

### 3.5 Accountable Offline Control with Belief Corpus

Given the linear relationship between the variables $v$ and $b$, for a specific action $a_t$, we can estimate the corresponding $\hat{v}(a_t) = \boldsymbol{l}(\hat{b}) = \sum_{c=1}^{C} w^c v^c$, where $o_t$ and $h_t$ are the control time observation and transition history respectively. Applying an arg-max type of operator to the above value estimator, we are able to select the action with the highest value that is supported by the belief corpus:

$$\pi_{\text{AOC}}(o_t, h_t) = \arg\max_{a_t} \hat{v}(a_t), \tag{8}$$

**Flexibility of the Measurement Function: Plug-and-Play.** As discussed earlier, the choice of measurement function $\mathbf{V}$ depends on the objective of policy learning. For the low-dimensional tasks, we can instantiate $\mathbf{V}$ using the Monte Carlo estimation of the cumulative return: $v^c = \sum_{t=t_c}^{T} \gamma^{t-t_c} r_t$, as the most straightforward choice in control tasks; for higher-dimensional tasks, the measurement metric can be the learned Q-value function; for other applications like when safety-constraints are also taken into consideration, the measurement function can be adapted accordingly.

### 3.6 Optimization Procedure

The optimization process in our proposed AOC method involves three main steps: optimizing the belief function and linear mapping, reconstructing the belief during control time rollout, and

---

**Algorithm 1** The Accountable Offline Controller

---

**Input**

    Bached Dataset $\mathcal{D} = \{o_t^i, a_t^i, r_t^i, o_{t+1}^i\}_{t=1,\dots,T}^{i=1,\dots,N}$, control time observation $o_t$, control history $h_t$. Hyper-parameters: $K$ for the number of candidate actions, $\epsilon$ for the threshold.

**Output**

    Accountable action $a_t$ for control.

\# 1. Optimize belief function $\boldsymbol{b}$ and linear mapping $\boldsymbol{l}$ according to Equation (9).

\# 2. Sample $K$ candidate actions uniformly in the action space: $a_t^k \sim U(\mathcal{A}), k \in [K]$.

\# 3. Find the minimal hulls $\tilde{\mathcal{C}}(b_t)$ for each action, based on $\boldsymbol{b}$ and $b_t = \boldsymbol{b}(o_t, a_t^k, h_t)$.

\# 4. Minimize corpus residuals for each action candidate $a_t^k$ according to Equation (10).

\# 5. Estimate the value of action $a_t$ by $\hat{v}(a_t|o_t, h_t) = \sum_{c=1}^C w^c v^c$

\# 6. Outputs the candidate action that has the highest estimated value according to Equation (11).

---

performing sampling-based optimization on the action space. In this section, we detail each of these steps along with corresponding optimization objectives. First, the belief function $\boldsymbol{b}$ and the linear mapping $\boldsymbol{l}$ are learned by minimizing the difference between the predicted performance measure and the actual value $v$ from the dataset $\mathcal{D}$:

$$\boldsymbol{b}, \boldsymbol{l} = \arg\min_{\boldsymbol{b}, \boldsymbol{l}} \mathbb{E}_{(o,a,h,v)\in\mathcal{D}} \left(v - \boldsymbol{l} \circ \boldsymbol{b}(o, a, h)\right)^2 \tag{9}$$

Next, with the observation $o_t$ and trace $h_t$, any candidate action $a_t$ corresponds to a belief $b_t = \boldsymbol{b}(o_t, a_t, h_t)$, which can be reconstructed by the minimal corpus subset $\tilde{\mathcal{C}}(b_t)$, according to:

$$w^c = \arg\min_{w^c} || \sum_{c=1}^C w^c \boldsymbol{b}(o^c, a^c, h^c) - b_t ||, \ \ \text{s.t.} \ \ \sum_{c=1}^C w^c = 1, \ (o^c, a^c, h^c) \in \tilde{\mathcal{C}}(b_t) \tag{10}$$

Then the value of action $a_t$ can be estimated by $\hat{v}(a_t|o_t, h_t) = \sum_{c=1}^C w^c v^c$. Finally, we perform a sampling-based optimization on the action space, with the corpus residual taken into consideration for the pursuance of conservation. For each control step, we uniformly sample $K$ actions in the action space $a^{(k)}t \sim U(\mathcal{A})$ and then search for the action that corresponds to the maximal performance measure while constraining the belief corpus residual to be smaller than a threshold $\epsilon \geq 0$:

$$\pi_{\text{AOC}}(o_t, h_t) = \arg\max_{a_t \in \{a_t^{(k)}\}} \sum_{c=1}^C w^c v^c, \quad \text{s.t.} \ \ r_{\mathcal{C}}(\boldsymbol{b}(o_t, a_t, h_t)) \leq \epsilon \tag{11}$$

The complete optimization procedure is detailed in Algorithm 1.

## 4 Related Work

Table 1: AOC is distinct as it satisfies **5 desired properties** mentioned earlier: (**P1**) Conservation: AOC seeks the best potential action in the belief convex hull, such that the estimations of decision outcomes are interpolated, avoiding aggressive extrapolation that is harmful in offline control [8]; (**P2**) Accountability: the decision-making process is supported by a corpus subset from the offline dataset, hence all the decisions are traceable; (**P3**) Low-Data Requirement: it works in the low-data regime; (**P4**) Adaptivity: the control behavior of AOC can be adjusted according to additional constraints as clinical guidelines without modification or re-training; (**P5**) Reward-Free: AOC can be extended to the strictly offline imitation setting where rewards are unavailable.

| Method / Property | Conservation | Accountable | Low-Data | Adaptive | Reward-Free | Examples |
|---|:---:|:---:|:---:|:---:|:---:|---|
| Q-Learning | ✓ | ✗ | ✓ | ✗ | ✗ | [8–16] |
| Episodic Control | ✗ | ✗ | ✓ | ✗ | ✗ | [17–19] |
| Nearest Neighbor | ✗ | ✓ | ✗ | ✓ | ✓ | [20, 21] |
| Model-Based RL | ✗ | ✗ | ✗ | ✓ | ✗ | [22–24] |
| Behavior Clone | ✗ | ✗ | ✗ | ✗ | ✓ | [25] |
| AOC | ✓ | ✓ | ✓ | ✓ | ✓ | Ours |

We review related literature and contrast their difference in Table 1. Due to constraints on space, we have included comprehensive discussions on related work in Appendix B. Our proposed method stands out due to its unique attributes: It incorporates properties of accountability, built-in conservation for offline learning, adaptation to preferences or guidelines, is suitable for the low-data regime, and is compatible with the strictly offline imitation learning setting where the reward signal is not recorded in the dataset.

# 5 Experiments

**Environments and Set-ups**    In this section, we present empirical evidence of the properties of our proposed method in a variety of environments. These environments include (1) control benchmarks that simulate POMDP and heterogeneous outcomes in healthcare, (2) a continuous navigation task that allows for the visualization of properties, and (3) a real-world healthcare dataset from Ward [26] that showcases the potential of AOC for deployment in high-stakes, real-world tasks.

In Section 5.1, we benchmark the performance of AOC on the classic control benchmark, contrasting it with other algorithms and highlighting **(P1-P3)**. In Section 5.2, Section 5.3, and Section 5.4 we individually highlight properties **(P1),(P2),(P4)**. Subsequently, in Section 5.5, we apply AOC to a real-world dataset, highlighting its ability to perform accountable, high-quality decision-making under the strictly offline imitation setting **(P1-P5)**. Finally, Section 5.6 provides additional empirical evidence supporting our intuitions and extensive stress tests of AOC.

## 5.1    **(P1-P3)**: Accountable Offline Control in the Low-Data Regime

**Experiment Setting**    Our initial experiment employs a synthetic task emulating the healthcare setting, simultaneously allowing for flexibility in stress testing the algorithms.

We adapt our environment, **Pendulum-Het**, from the classic Pendulum control task involving system stabilization. Heterogeneous outcomes in healthcare contexts are portrayed through varied dynamics governing the data generation process. Data generation process details can be found in Appendix D.4. Taking heterogeneous potential outcomes and partial observability into consideration, the Pendulum-Het task is significantly more challenging than the classical counterpart.

We compare AOC with the nearest-neighbor controller (**1NN**) [20] and its variant that using $k$ neighbors (**kNN**), model-based RL with Mode Predictive Control (**MPC**) [24], model-free RL (**MFRL**) [27], and behavior clone (**BC**) [25]. Our implementation for those baseline algorithms is based on open-sourced codebases, with details elaborated in Appendix D.6. We change the size of the dataset to showcase the performance difference of various methods under different settings. The **Low-Data** denotes settings with 100000 transition steps, **Mid-Data** denotes settings with 300000 transition steps, **Rich-Data** denotes settings with 600000 transition steps.

**Results**    We compare AOC against baseline methods in Table 2. AOC achieves the highest average cumulative score among the accountable methods and is comparable with the best black-box controller. In experiments, we find increasing the number of offline training examples does not improve the performance of MFRL due to stability issues. Detailed discussions can be found in Appendix D.7.

Table 2: Results on the Heterogeneous Pendulum dataset. The cumulative reward of each method is reported. Experiments are repeated with 8 seeds. **Higher is better.**

| Task | Low-Data | Mid-Data | Rich-Data |
|---|---|---|---|
| AOC | $\mathbf{-1.39 \pm 1.39}$ | $\mathbf{-1.25 \pm 0.40}$ | $\mathbf{-0.6 \pm 0.08}$ |
| kNN | $-849.45 \pm 91.23$ | $-670.51 \pm 321.09$ | $-645.72 \pm 220.33$ |
| 1NN | $-557.07 \pm 256.64$ | $-690.49 \pm 152.59$ | $-512.71 \pm 131.2$ |
| BC | $-422.77 \pm 409.51$ | $-225.32 \pm 340.83$ | $-126.74 \pm 280.73$ |
| MFRL | $-4.1 \pm 2.76$ | $-11.95 \pm 4.68$ | $-15.27 \pm 6.46$ |
| MPC | $\mathbf{-1.5 \pm 0.43}$ | $\mathbf{-1.34 \pm 0.15}$ | $-1.41 \pm 0.26$ |
| Data-Avg-Return | $-307.81 \pm 387.53$ | $-245.54 \pm 338.65$ | $-208.81 \pm 272.84$ |

**Take-Away:** *As an accountable algorithm, AOC performs on par with state-of-the-art black-box learning algorithms. Particularly in the POMDP task, wherein data originates from heterogeneous systems, AOC demonstrates robustness across diverse data availability conditions.*

## 5.2    **(P1)**: Avoid Aggressive Extrapolation with the Minimal Hull and Minimized Residual

**Experiment Setting**    In Equation (11), our arg-max operator includes two adjustable hyperparameters: the number of uniformly sampled actions and the threshold. In this section, we demonstrate how these hyperparameters can be unified and easily determined.

The consideration of sample size primarily represents a balance between optimization precision and computational cost — although such cost can be minimized by working with modern computational platforms through parallelization. On the other hand, selecting an appropriate value for $\epsilon$ might initially seem complex, as the residual's magnitude should depend on the learned belief space.

In our implementation, we employ a quantile number to filter out sampled actions whose corresponding corpus residuals exceed a certain population quantile. For instance, when $\epsilon = 0.3$ with a sample size of 100, only the top $30\%$ of actions with the smallest residuals are considered in the arg-max operator in Equation (11). The remaining number of actions is referred to as the *effective action size*. This section presents results for the **Pendulum-Het** environment, while additional qualitative and quantitative results on the conservation are available in Appendix F.2. Intuitively, using a larger sampling size will improve the performance, at a sub-linear cost of computational expenses, and using a smaller quantile number will make the behavior more conservative. That said, using a small quantile number may not be always good, as it also decreases the effective action size. The golden rule is to use larger sampling size and smaller quantile threshold.

**Results**  Table 3 exhibits the results. Our experiments highlight the significance of this filtering mechanism: removal or weakening of this mechanism, through a larger quantile number that filters fewer actions, leads to aggressive extrapolation and subsequently, markedly deteriorated performance. This is because the decisions made by AOC in those cases cannot be well supported by the minimal convex hull, introducing substantial epistemic uncertainty in value estimation.

Table 3: The $\epsilon$ in Equation (11) controls the conservation tendency of the control time behaviors.

| $\epsilon$ (quantile) | Performance |
|---|---|
| 0.3 | $-2.05 \pm 0.45$ |
| 0.5 | $-1.25 \pm 0.40$ |
| 0.9 | $-503.39 \pm 301.60$ |
| 1.0 | $-853.29 \pm 138.43$ |

Conversely, using a smaller quantile number yields a reduced *effective action size*, which subsequently decreases optimization accuracy and leads to a minor performance drop. Hence when computational resources allow, utilizing a larger sample size and a smaller quantile number can stimulate more conservative behavior from AOC. In all experiments reported in Section 5.1, we find a sample size of 100 and $\epsilon = 0.5$ generally work well.

**Take-Away:** *AOC is designed for offline control tasks by adhering to the principle of conservative estimation. The $\epsilon$ hyperparameter, introduced for conservative estimation, can be conveniently determined by incorporating it into the choice of an effective action size.*

### 5.3  **(P2): Accountable Offline Control by Tracking Reference Examples**

**Experiment Setting**  In healthcare, the treatments suggested by the clinician during different phases can be based on different patients, and those treatments may even be given by different clinicians according to their expertise. In this section, we highlight the accountability of AOC within a **Continuous Maze** task. Such a task involves the synthesis of decisions from multiple experts to achieve targeted goals. It resembles the healthcare example above and is ideal for visualizing the concept of accountability.

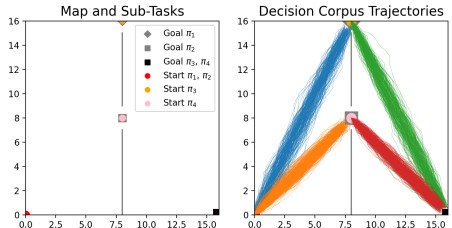

Figure 2: Left: the Maze; Right: trajectories generated by 4 experts for different tasks.

Figure 2 displays the map, tasks, and expert trajectories within a 16x16 Maze environment. This environment accommodates continuous state and action spaces. We gathered trajectories from 4 agents, each of which is near-optimal for a distinct navigation task:
① agent $\pi_1$ starts at position $(0,0)$ and targets position $(8,16)$. Trajectories of $\pi_1$ are marked in blue.
② agent $\pi_2$ starts at position $(0,0)$ and targets position $(8,8)$. Trajectories of $\pi_2$ are marked in orange.
③ agent $\pi_3$ starts at position $(8,16)$ and aims position $(16,0)$. Trajectories of $\pi_3$ are marked in green.
④ agent $\pi_4$ starts at position $(8,8)$ and targets position $(16,0)$. Trajectories of $\pi_4$ are marked in red.

We generate 250 trajectories for each of the 4 agents, creating the offline dataset. The control task initiates at position $(0,0)$, with the goal situated at $(16,0)$ — requires a fusion of expertise in the dataset to finish. To complete the task, an agent could potentially learn to replicate $\pi_1$, followed by adopting $\pi_3$'s strategy. Alternatively, it could replicate $\pi_2$ before adopting $\pi_4$'s approach.

**Results**   In this experiment, we employ AOC to solve the task and visualize the composition of the corpus subset at each decision step in Figure 3. In the initial stages, the corpus subset may comprise a blend of trajectories from $\pi_1$ and $\pi_2$. Subsequently, AOC selects one of the two potential solutions at random, as reflected in the chosen reference corpus subsets. Upon crossing the central gates on the x-axis, AOC's decisions become reliant on either $\pi_3$ or $\pi_4$, persisting until the final steps, when the corpus subsets can again be a mixture of the overlapped trajectories.

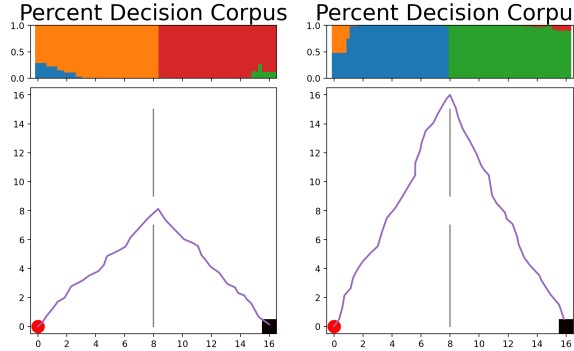

Figure 3: Visualize Accountability. The upper colors display the components of the corpus subset at different x-positions.

**Take-Away:** *AOC's decision-making process exhibits strong accountability, as the corpus subset can be tracked at every decision-making step. This is evidenced by AOC's successful completion of a multi-stage Maze task, wherein it effectively learns from mixed trajectories generated by multiple policies at differing stages.*

## 5.4   (P4): Flexible Offline Control under User Specification

**Experiment Setting**   Given the tractable nature of AOC's decision-making process, the agent's behavior can be finely tuned. One practical way of modifying these decisions is to adjust the reference dataset, which potentially contributes to the construction of the corpus subset. In this section, we perform experiments in the **Continuous Maze** environment with the same offline dataset described in Section 5.3. We modify the sampling rate of trajectories generated by $\pi_1$ and experiment with both re-sampling and sub-sampling strategies, using sample rates ranging from $[\times 4, \times 3, \times 2, \times 1, \times 0.75, \times 0.5, \times 0.25]$. In the re-sampling experiments (sample rates larger than 1), we repeat trajectories produced by $\pi_1$ to augment their likelihood of selection as part of the corpus subset. In contrast, the sub-sampling experiments partially omit trajectories from $\pi_1$ to reduce their probability of selection. In each setting, 100 control trajectories are generated.

**Results**   Figure 4 shows the results: the success rate (**Succ. Rate**) quantifies the quality of control time behaviors, while the remaining two curves denote the percentage of control time trajectories that accomplish the task by either following either $\pi_1$ - $\pi_3$ (passing the upper gate) or $\pi_2$ - $\pi_4$ (passing the lower gate). The behavior of AOC is influenced by the sampling rate. Utilizing a high sampling rate for the trajectories generated by $\pi_1$ often leads to the selection of the solution that traverses the upper gate. As the sampling rate diminishes, AOC's behavior begins to be dominated by the alternative strategy. In both re-sampling

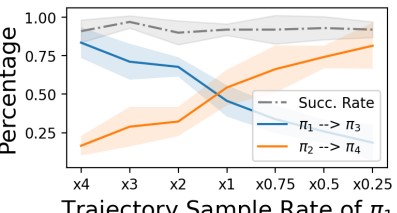

Figure 4: The behavior of AOC can be controlled by re-sampling or sub-sampling the offline dataset.

and sub-sampling strategies, the success rate of attaining the goal remains consistently high. We provide further visualizations of the learned policies under various sampling rates in Appendix E.

**Take-Away:** *The control time behavior of AOC can be manipulated by adjusting the proportions of behaviors within the offline dataset. Both re-sampling and sub-sampling strategies prove equally effective in our experiments where multiple high-performing solutions are available.*

## 5.5   (P1-P5): Real-World Dataset: The Strictly Batch Imitation Setting in Healthcare

**Experiment Setting**   We experiment with the **Ward dataset** [26] that is publicly available with [28]. The dataset contains treatment information for more than 6000 patients who received care on the general medicine floor at a large medical center in California. These patients were diagnosed with a range of ailments, such as pneumonia, sepsis, and fevers, and were generally in stable condition. The measurements recorded included standard vital signs, like blood pressure and pulse, as well as lab test results. In total, there were 8 static observations and 35 temporal observations that captured the dynamics of the patient's conditions. The action space was restricted to a choice of a binary

action pertaining to the use of an oxygen therapy device. In the healthcare setting, treatment decisions are given by domain experts and the objective of policy learning is to mimic those expert behaviors through imitation. However, as interaction with the environment is strictly prohibited, the learning and evaluation of policies should be conducted in a strictly offline manner. The dataset is split into $10 : 1$ training and testing sets and the test set accuracy is used as the performance metric.

We benchmark the variant of AOC tailored for the strictly batch imitation setting, dubbed as **ABC** — as an abbreviation for **A**ccountable **B**ehavior **C**lone — against three baselines: (1) behavior cloning with a linear model (**BC-LR**), which serves as a transparent baseline in the offline imitation setting; (2) behavior cloning with a neural network policy class (**BC-MLP**), which represents the level of performance achievable by black-box models in such contexts; and (3) the k-nearest neighbor controller (**kNN**) as an additional accountable baseline.

**Results** The results of kNN with various choices of $k$ in the experiments, alongside a comparison with ABC using the same number of examples in the corpus subset, are presented in Figure 5. Shaded areas in the figure denote the variance across 10 runs.

Operating as an accountable policy under the offline imitation setting, ABC achieves performance similar to that of behavior cloning methods utilizing black-box models. Meanwhile, other baselines face challenges in striking a balance between transparency and performance.

Furthermore, we visualize the corpus subset supporting control time decisions in a 2-dimensional belief space to illustrate the accountability. We demonstrate the differences in the decision boundary where predictions are more challenging. In such cases, the decision supports of kNN rely on extrapolation and have no idea of the potential risk of erroneous prediction, while the heterogeneity in ABC's minimal convex hull reflects the risk. We provide a more detailed discussion in Appendix E.2.

**Take-Away:** *The variant of AOC, dubbed as ABC, is well-suited for the strictly batch imitation setting, enabling accountable policy learning. While other baseline methods in this context struggle to balance transparency and policy performance, ABC achieves comparable performance with black-box models while maintaining accountability.*

Figure 5: Results on the Ward dataset. ABC is able to perform accountable imitation under the strict offline setting with high performance comparable with the black-box models.

### 5.6 Additional Empirical Studies

To further verify our intuitions and stress test AOC, we provide additional empirical analysis in Appendix F. Specifically, we analyze the **trade-off** between accountability and performance in Appendix F.1; we visualize the **controllable conservation** in Appendix F.2; we benchmark AOC on **another benchmark** for the general interests of the RL community in Appendix F.3; and show how to combine AOC with **black-box sampler** in Appendix F.4; Finally, we show AOC can detect control time **OOD examples** in Appendix F.5.

## 6 Conclusion

In conclusion, this study presents the Accountable Offline Controller (AOC) as a solution for offline control in responsibility-sensitive applications. AOC possesses five essential properties that make it particularly suitable for tasks where traceability and trust are paramount: ensures accountability through offline decision data, thrives in low-data environments, exhibits conservative behavior in offline contexts, can be easily adapted to user specifications, and demonstrates flexibility in strictly offline imitation learning settings. Results from multiple offline control tasks, including a real-world healthcare dataset, underscore the potential of AOC in enhancing trust and accountability in control systems, paving the way for its widespread adoption in real-world applications.

## Acknoledgement

HS acknowledges and thanks the funding from the Office of Naval Research (ONR). HS would thank Jonathan Crabbe for his inspiration and encouragement in exploring this idea in a group meeting. We thank Jonathan Crabbe for reviewing our manuscript and providing insightful comments and suggestions for improving the paper. We thank all anonymous reviewers, ACs, SACs, and PCs for their efforts and time in the reviewing process and in improving our paper. This work is done with the warm support from the van der Schaar Lab members.

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

# Accountability in Offline Reinforcement Learning: Explaining Decisions with a Corpus of Examples

## Appendix: Table of Contents

# A    Missing Proofs

## A.1    Proof of Proposition 3.8 (See page 4)

**Proposition 3.8** (Estimation Error Bound for $v(a_t)$, *cf.* Crabbé et al. [7]). *Consider the belief variable* $b_t = \boldsymbol{b}(o_t, a_t, h_t)$ *and* $\hat{b}$ *the optimizer of Equation (6), the estimated value residual between* $\boldsymbol{l}(b_t)$ *and* $\boldsymbol{l}(\hat{b})$ *is controlled by the corpus residual:*

$$||\boldsymbol{l}(\hat{b}) - \boldsymbol{l}(b_t)||_{\mathcal{V}} \leq ||\boldsymbol{l}||_{\mathrm{op}} \cdot r_{\mathcal{C}}(b_t) \tag{7}$$

*where* $|| \cdot ||_{\mathcal{V}}$ *is a norm on* $\mathcal{V}$ *and* $||\boldsymbol{l}||_{\mathrm{op}} = \inf \{\lambda \in \mathbb{R}^+ : ||\boldsymbol{l}(b)||_{\mathcal{V}} \leq \lambda ||b||_{\mathcal{B}}\}$ *is the operator norm for the linear mapping.*

*Proof.* Leveraging the linearity of operator $\boldsymbol{l}$, the definition of the operator norm $|| \cdot ||_{\mathrm{op}}$, and Definition 3.7, we have:

$$\begin{aligned}
||\boldsymbol{l}(\hat{b}) - \boldsymbol{l}(b_t)||_{\mathcal{V}} &= ||\boldsymbol{l}(\hat{b} - b_t)||_{\mathcal{V}} \\
&\leq ||\boldsymbol{l}||_{\mathrm{op}} \cdot ||\hat{b} - b_t||_{\mathcal{V}} \\
&= ||\boldsymbol{l}||_{\mathrm{op}} \cdot r_{\mathcal{C}}(b_t)
\end{aligned} \tag{12}$$

$\square$

## A.2    Proof of Proposition 3.10 (See page 5)

**Lemma A.1** (Affine Independence of $\tilde{\mathcal{C}}(b_t)$). *The elements in the belief corpus built on top of* $\tilde{\mathcal{C}}(b_t)$, *as the corpus subset:* $b^c \in \boldsymbol{b}(\tilde{\mathcal{C}}(b_t)) \subset \mathcal{B}$ *must be affinely independent, that is*

$$\sum_{c=1}^{C} \lambda^c b^c = 0 \wedge \sum_{c=1}^{C} \lambda^c = 0 \implies \lambda^c = 0, \forall c \in [C] \tag{13}$$

*Proof.* The proof is based on contradiction. Note that by definition, there are $d_b + 1$ elements in the belief corpus set. If those elements are not affinely independent, it basically means that $b_t$ can be expressed in a lower dimensional space. In this case, the composition of the convex set is redundant. Without loss of generality, we use $b^1$ to denote the redundant element

$$b^1 = -\sum_{c=2}^{C} \frac{\lambda^c}{\lambda^1} b^c, \tag{14}$$

indicating that without $b^1$, we still have $b_t \in \mathcal{CB}(\mathcal{C}')$, where $\mathcal{C}' = \tilde{\mathcal{C}}(b_t) \backslash (o^1, a^1, h^1)$. This contradicts that $\tilde{\mathcal{C}}(b_t)$ is the minimal hull that contains $b_t$. $\square$

**Proposition 3.10** (Existence and Uniqueness). *Consider the belief variable* $b_t = \boldsymbol{b}(o_t, a_t, h_t)$, *if* $r_{\mathcal{C}}(b_t) \leq 0$ *holds for* $\mathcal{C} = \mathcal{D}$, *then* $\tilde{\mathcal{C}}(b_t)$ *exists and the decomposition on the corresponding minimal convex hull exists and is unique.*

*Proof.* Following the assumption, there is at least one trivial corpus subset that exists for $b_t$ — the offline dataset $\mathcal{D}$ as the corpus subset — such that a zero belief corpus residual can be achieved. [2] The existence of $\tilde{\mathcal{C}}(b_t)$ follows the fact the cardinality of the $\mathcal{D}$ is a finite number. The existence of decomposition on $\tilde{\mathcal{C}}(b_t)$ is then a consequence of the definition of the minimal hull.

Based on Lemma A.1, the elements in $\tilde{\mathcal{C}}(b_t)$ construct the minimal hull and are affinely independent. The uniqueness of the decomposition can be shown by contradiction:

Assume that there are two different decompositions for composing the belief state $b_t$:

$$b_t = \sum_{c=1}^{C} \omega^c b^c = \sum_{c=1}^{C} \tilde{\omega}^c b^c, \tag{15}$$

---

[2]otherwise, it falls into an out-of-distribution prediction problem and is beyond the scope of this work. Nonetheless, we note that AOC is able to perform OOD detection. See section F.5

where $\omega^c, \tilde{\omega}^c \in [0, 1], \sum_{c=1}^{C} \omega^c = 1, \sum_{c=1}^{C} \tilde{\omega}^c = 1$. In this case, we have

$$
\begin{aligned}
0 &= \sum_{c=1}^{C} \omega^c b^c - \sum_{c=1}^{C} \tilde{\omega}^c b^c \\
&= \sum_{c=1}^{C} (\omega^c - \tilde{\omega}^c) b^c \\
&= \sum_{c=1}^{C} \lambda^c b^c, \text{where } \lambda^c \equiv \omega^c - \tilde{\omega}^c.
\end{aligned}
\tag{16}
$$

However, $\sum_{c=1}^{C} \lambda^c = \sum_{c=1}^{C} (\omega^c - \tilde{\omega}^c) = \sum_{c=1}^{C} \omega^c - \sum_{c=1}^{C} \tilde{\omega}^c = 1 - 1 = 0$ contradicts with the fact that the elements are affinely independent. Hence the proof is completed. $\square$

## B Extended Related Work

### B.1 Offline RL

Offline RL [8–14, 29, 30, 15, 16] has gained increasing attention in recent years due to its potential for solving practical problems, such as robotics control and game playing, where collecting new data can be expensive or time-consuming. However, it also presents several challenges, such as distribution shift and overfitting, which can lead to poor performance when deploying the learned policy to the actual environment. There are several model-free approaches to addressing these challenges in Offline RL. Such as distributional matching [8, 31], regularization techniques to prevent overfitting [32], conservation [12, 33] or adding noise [15] to the policy or using adversarial training [34]. A third approach is to incorporate uncertainty estimation to evaluate the performance of the learned policy on unseen data [13].

On the other hand, model-based RL tackles the problem by first learning world models and then performing planning algorithms on the learned model [22, 23, 35, 36]. In either model-based or model-free approaches, black-box approximators are used; hence, the decision is not transparent.

In offline-RL, both model-based and model-free approaches leverage black-box approximators. As a consequence, the pursuance of accountability can not be achieved through those conventional algorithms.

We would like to note that, although AOC also studies the control problems under the offline setting, its focus goes beyond the conservative efficient learning objective in offline-RL literature. As we have demonstrated in the Table 1, AOC has five distinct properties that are all crucial for accountable offline control tasks:

- (P1). Conservation.
- (P2). Accountability.
- (P3). Low-Data Requirement.
- (P4). Adaptivity.
- (P5). Reward-Free.

Below, we further explain those properties and corresponding methods in turn. For each of the properties, we start with introducing the definitions, followed by comparisons among AOC and MFRL, MBRL, and BC. The discussion on MFRL and MBRL includes the offline-RL algorithms.

(1) Accountability: the decision-making process is traceable, and the decisions can be supported by concrete examples in the offline dataset.

- AOC: The decision-making process of AOC is supported by a corpus subset from the offline dataset, hence all the decisions are transparent and traceable.
- MFRL: In MFRL, a black-box value network and black-box policy network are learned with the offline dataset. There is no decision support for the black-box policies.
- MBRL: In MBRL, a black-box world model optimized with the offline dataset is used as a proxy of the actual dynamics, and planning algorithms are then applied to such a black-box model to make decisions. Those decisions are not supported by explicit references.

- BC: In BC, a black-box policy is learned through supervised learning. The output of such a policy is hard to be linked with specific training examples.

(2) Conservation: estimations of decision outcomes are interpolated, avoiding aggressive extrapolation that is harmful in offline control.

- AOC: AOC performs conservative decision-making by using decision supports within a minimal convex hull. How such a decomposition in the convex hull improves conservation is justified theoretically by Proposition 3.8 (Estimation Bound) and Proposition 3.10 (Existence and Uniqueness).
- MFRL: In MFRL like CQL and TD3-BC, the conservation is explicitly given as constraints or distribution matching. We would note that conventional MFRL algorithms are not designed for those tasks and suffer from aggressive extrapolation.
- MBRL: Similar to MFRL, external efforts should add conservation to MBRL. Because the conventional design of model-based learning does not address such an issue.
- BC: In BC, the learning objective is to minimize the prediction difference. There is little we can do to aid conservation.

(3) Low-Data: whether a method works in the low-data regime.

- AOC: The decision process of AOC only relies on a few examples forming the minimal convex hull, hence the algorithm performs well under the low-data regime, making it generally applicable to many real-world data-scarce tasks.
- MFRL: In MFRL, the black-box value network and policy network can be designed to be sample-efficient.
- MBRL: In MBRL, sufficient data is always required to learn an accurate world model.
- BC: the performance of BC is highly dependent on the data quality. It is not designed for the low-data regime.

(4) Adaptive: whether the control behavior of a method can be adjusted according to additional constraints as clinical guidelines without modification or re-training.

- AOC: by changing reference examples, i.e., the decision corpus, during test time inference, AOC can seamlessly perform different types of decision-making according to user specifications.
- MFRL: In MFRL, when new data is used, a new value network and policy network need to be re-trained.
- MBRL: In MBRL, the world model construction is independent of the data, hence the decisions can be adaptive by changing a new planning algorithm on top of the world model. No model re-training is needed.
- BC: In BC, a new model needs to be trained with a specified type of decision corpus.

(5) Reward-Free: availability of extension to the strictly batched imitation setting where rewards are unavailable.

- AOC: AOC can be extended to the strictly batched imitation setting where rewards are unavailable.
- MFRL: In MFRL, the Q-values can not be calculated without the reward function.
- MBRL: In MBRL, the planning algorithms do not have a clear objective to optimize without reward signals.
- BC: BC is not affected by the absence of reward signals, because it does not need the reward to learn its policy.

## B.2 Episodic Control and Nearest Neighbor Control

Episodic Memory and Episodic Control [17–19], inspired by biological learning mechanism, are studied as an alternative way of policy learning [37]. Follow-up works introduce various modifications and extend episodic control to the continuous domains. e.g., Hu et al. [38] introduces Generalized Episodic Memory (GEM) which effectively organizes the state-action values of episodic memory in a generalizable manner and supports implicit planning on memorized trajectories. Ma et al. [39] leverages episodic memory in offline RL setting with a pessimistically estimated value function. Li et al. [40] proposing a novel state-abstractor framework for episodic control and improving the learning efficiency in continuous control benchmarks. In all those works, a value function

resembling the hippocampus episodic memorization mechanism is introduced as an alternative to Q-learning [41, 42] that updates the state-action values with temporal difference learning or Monte-Carlo estimation [43–45].

In the continuous control domains, policy gradient methods [46, 47] or supervised learning-based methods [48, 49, 10, 50] are then applied for the policy improvement. All of those approaches are limited to black-box value-based learning and require additional black-box policy networks in the continuous control domain, whereas our proposed method performs a transparent decision-making process without explicit policy learning.

Explicit nearest neighbor methods that perform decision-making according to training-time similar trajectories have been studied theoretically [20] and empirically [21]. Although those methods also enjoy transparency, they suffer from the problem of the curse of dimensionality. Moreover, defining the nearest neighbor with a heuristically determined Euclidean metric also suffers the problem of aggressive extrapolation and thus is not suitable for the offline setting [8, 12].

### B.3 Explainable RL

Understanding the decisions made by RL agents is a key issue in many high-stake real-world domains such as autonomous driving [51], finance [52, 53] and healthcare [54, 28, 55–59]. Previous Explainable-RL (XRL) literature can be broadly classified with a taxonomy of three classes [60]: (1) Feature importance, that includes learning policy through an explainable policy class [61–63], converting black-box models into an interpretable format [64–68], and natural language [69–71] or saliency map based explanations [72–74]; (2) Transparent Learning Process that reveals the influences of MDP ingredients during the learning process, including methods that learn to predict the counterfactual outcomes for decision-making [75–77], decompose the learning objective [78–81], and identifying the crucial training datum [82, 83]; (3) Policy Level, which illustrates the long-term behavior of the agent [84–86]. For more extensive discussions on XRL literature, we refer the readers also to [87, 88]. Different from previous approaches, our work introduces the first example-based explanation for policy learning. Supported by training dataset examples, the execution of our control algorithm is accountable.

## C The Strictly Batch(Offline) Imitation Setting

The instant reward may not be contained in the offline dataset in strictly batch imitation (SBI) [31] learning settings such as clinical treatment and healthcare scenarios. In such cases, the value of actions can not be estimated through Monte Carlo, and it is generally impossible to learn a belief state based on the value prediction. In such a case, we need to adapt the Accountable Offline Controller accordingly.

In such a setting, the dataset $\mathcal{D}_{\text{SBI}} = \{o_t^i, a_t^i, o_{t+1}^i\}_{t=1,...,T}^{i=1,...,N}$ contains only sequential observations $o_t^i, o_{t+1}^i, i \in [N], t \in [T]$, actions $a_t^i, i \in [N], t \in [T]$ performed by behavior policies, which is always an expert or near-expert controller, and transition histories $h_t^i, i \in [N], t \in [T]$ that can be composed of the former quantities.

The Accountable Offline Controller should be adapted to handle this setting. Specifically, we can still define the corpus subset as

**Definition C.1** (SBI Corpus Subset). A *SBI Corpus Subset* $\mathcal{C}_{\text{SBI}}$ is defined as a subset of an offline dataset $\mathcal{D}_{\text{SBI}}$, indexed by $[C] := \{1, 2, ..., C\}$ — the natural numbers between $1$ and $C$.

$$\mathcal{C}_{\text{SBI}} = \left\{ (o^c, a^c, h^c) \in \mathcal{D}_{\text{SBI}} \Big| c \in [C] \right\}. \tag{17}$$

**Property C.2.** (SBI Linear Restricted Belief) The policy function $\boldsymbol{\pi} : \mathcal{O} \times \mathcal{H} \mapsto \mathcal{A}$ can be decomposed as $\boldsymbol{\pi} = \boldsymbol{l} \circ \boldsymbol{b}$, where $\boldsymbol{b} : \mathcal{O} \times \mathcal{H} \mapsto \mathcal{B} \subseteq \mathbb{R}^{d_b}$ maps the joint observation-history space to a $d_b$ dimensional belief variable $b_t = \boldsymbol{b}(o_t, h_t)$ and $\boldsymbol{l} : \mathcal{B} \mapsto \mathcal{A}$ is a linear function that maps the belief $b_t$ to an output $\boldsymbol{l}(b_t) \in \mathcal{A}$.

Then any control time behavior generated by policy $\pi$ is accountable in the sense that it can be decomposed by the belief corpus, defined as

**Definition C.3.** (SBI Belief Corpus) A *SBI Belief Corpus* $\boldsymbol{b}(\mathcal{C}_{\mathrm{SBI}})$ is defined by applying the belief function $\boldsymbol{b}$ to the corpus subset $\mathcal{C}_{\mathrm{SBI}}$,

$$\boldsymbol{b}(\mathcal{C}_{\mathrm{SBI}}) = \left\{ b^c = \boldsymbol{b}(o^c, h^c) \Big| (o^c, a^c, h^c) \in \mathcal{C}_{\mathrm{SBI}} \right\} \subset \mathcal{B}, \tag{18}$$

on top of which we can define the Belief Corpus Hull in the SBI setting:

**Definition C.4.** (SBI Belief Corpus Hull) The *SBI Belief Corpus Convex Hull* spanned by a corpus subset $\mathcal{C}_{\mathrm{SBI}}$ with the belief corpus $\boldsymbol{b}(\mathcal{C}_{\mathrm{SBI}})$ is the convex set

$$\mathcal{CB}(\mathcal{C}_{\mathrm{SBI}}) = \left\{ \sum_{c=1}^{C} w^c b^c \Big| w^c \in [0,1], b^c \in \boldsymbol{b}(\mathcal{C}_{\mathrm{SBI}}), \forall c \in [C], \sum_{c=1}^{C} w^c = 1 \right\}, \tag{19}$$

followed by the concept of Minimal Hull in the SBI setting defined as

**Definition C.5** (SBI Minimal Hull). Denoting the decision corpora whose belief convex hull contain a belief variable $b_t$ by

$$\mathcal{C}_{\mathrm{SBI}}(b_t) = \left\{ \mathcal{C}_{\mathrm{SBI}} \subseteq \mathcal{D}_{\mathrm{SBI}} \Big| b_t \in \mathcal{CB}(\mathcal{C}_{\mathrm{SBI}}) \right\}$$

Among those subsets, the one that contains $d_b + 1$ decision corpora and has the smallest hyper-volume forms the *minimal hull*, denoted by $\tilde{\mathcal{C}}_{\mathrm{SBI}}(b_t)$:

$$\tilde{\mathcal{C}}_{\mathrm{SBI}}(b_t) := \min_{\mathcal{C}_{\mathrm{SBI}}} \int_{\mathcal{CB}(\mathcal{C}_{\mathrm{SBI}})} dV$$

Similar to the property in AOC, in the SBI setting, the control time decisions can be decomposed with examples in the offline dataset that constructs such an SBI Minimal Hull,

$$\pi(a_t|o_t, h_t) = \boldsymbol{l} \circ \boldsymbol{b}(o_t, h_t) = \boldsymbol{l} \circ \sum_{c=1}^{C} w^c \cdot \boldsymbol{b}(o^c, h^c) = \sum_{c=1}^{C} w^c \cdot \boldsymbol{l} \circ \boldsymbol{b}(o^c, h^c) = \sum_{c=1}^{C} w^c \cdot a^c,$$

where $(o^c, a^c, h^c) \in \tilde{\mathcal{C}}_{\mathrm{SBI}}(b_t)$.

# D    Further Implementation and Experiment Details

## D.1    Reproduceability: Code

We elaborate on our implementation details in this section. Our code is available at `https://github.com/orgs/vanderschaarlab/repositories/accountableofflinerl`.

## D.2    Learning the Belief Space

To efficiently encode the information in historical transition, in our work, we employ the Gated Recurrent Units (GRU) [89] to map fixed-length transition histories into embedding vector variables, followed by 3 fully connected layers as the belief function $\boldsymbol{b}$. In principle, any other recurrent networks should also be able to process such context information. Table 4 presents the hyper-parameters we use in the belief learning process.

## D.3    Addressing the Scalability Issue: Finding the Minimal Convex Hull Efficiently

Rigorously searching for the minimal convex hull in the belief space is a combinatorial optimization problem. In our implementation, we leverage a heuristic search method that first reduces the search space by looking for $k$-nearest neighbors in the belief space, and then build the approximate minimal convex hull on top of those $k$-nearest neighbors of the control time belief $b_t$. With $d_b$-dimensional belief space, the minimal convex hull will contain at most $d_b + 1$ examples, hence we can set $k = 2(d_b + 1)$, and conduct the combinatorial optimization on those $k$ examples, which is much easier than the original problem (reduce combinatorial problem *Select $d_b + 1$ out of $N$* into *Select $d_b + 1$ out of $k$*).

Table 4: Hyperparameters in learning the belief function.

| Hyper-Param | Choice |
|---|---|
| Context Model | GRU |
| Hidden Unit Number | 128 |
| Hidden Recurrent Layer | 1 |
| Batch Size | 500 |
| Epochs | 4000 |
| Optimizer | Adam |
| Learning Rate | 0.001 |
| Memory Length | 4 |
| Embedding Dimension | 20 |

## D.4  Data Generation Process: Heterogeneous Pendulum

The classical control task of Pendulum has the goal to swing up and balance a pendulum using a control input. In the control task, a policy can apply torque to the joint in order to swing the pendulum up and then maintain its upright position. The state of the system is defined by the pendulum's angle and angular velocity, and the action is the torque applied to the joint. The reward function typically provides a positive reward for keeping the pendulum upright and a negative reward for large torques or deviations from the upright position.

To manifest the potential heterogeneous outcome in healthcare and generalize the study into POMDP, we consider a heterogeneous variant of the original task. There are two contradictory Pendulum systems: the normal one and the converse one. While in the first system, adding torque will lead to a dynamical change according to the original physical design, in the converse system, the torque inputs will be sent to the system with a negation.

We train TD3 policies in each system and merge the collected dataset together as the offline dataset. In the Low-Data settings, 50000 transitions of each environment are collected, hence the dataset contains in total of 100000 transitions; in the Mid-Data regime, 150000 transitions of each environment are collected, hence the dataset contains in total of 300000 transitions; in the Rich-Data regime, 300000 transitions of each environment are collected, hence the dataset contains in total of 600000 transitions.

In order to achieve high performance, the agent must be able to identify the decision corpora that are collected from the same system dynamics as in the control time.

## D.5  Hardware and Running Time

We experiment on a machine with 2 TITAN X GPUs and 32 Intel(R) E5-2640 CPUs. In general, the computational expense of model training in AOC is cheap, as the neural networks used in AOC are in general shallow and of small scale. Learning the belief space requires half the calculation of building a world model. However, we acknowledge the exact calculation of the minimal convex hull in a large-data regime can be computationally expensive. And we have discussed our practical solution (Appendix D.3). With our proposed solution, the convex hull decomposition takes less than 10 seconds with a uniform sampler that samples 100 actions randomly for every time step. Increasing the dimension of belief space size will lead to a sub-linear increase in computational time with parallelization.

## D.6  Baseline Implementations

**Benchmark Algorithms**   Except for standardized components that we will introduce below, we use the publicly available source code when constructing the benchmark algorithms; references are in the following:

- **kNN and 1NN**: https://scikit-learn.org/.../neighbors, Reference: [20].
- **BC**: Implementation is straightforward using supervised learning.
- **MFRL**: https://github.com/sfujim/TD3, Reference: [27].
- **MPC**: https://github.com/UM-ARM-Lab/pytorch-mppi/.../mppi.py, Reference: [90].

**Neural Network Backbones** In all baseline methods and our implementation for AOC, we use the same network structure: 3-layer MLP with a recurrent model that encodes the historical trajectory information, which is called as the *Context Variable* in the literature [91–94]. Our implementation of the recurrent model is based on the open-sourced code of https://github.com/amazon-science/meta-q-learning. In all experiments, we use the same neural network architecture and match the hyper-parameters for a fair comparison, except stated otherwise (e.g., the Q-learning baseline requires a larger batch size to guarantee convergence and boost stability, which will be elaborated in the following section.).

### D.7 The Model-Free RL Baseline and Its Stability Issue

Our implementation of the Q-learning baseline leverages the twin-delayed techniques [27] to stabilize training. We find that Q-learning requires a large batch size (i.e., 10240) and many optimization epochs to converge. In the experiment settings with more offline data, the convergence becomes even harder: the rich-data regime containing $6\times$ more data takes $6\times$ more training epochs to converge. And the converged performance is always with high vibration, leading to worse performance.

Learning curves are shown in Figure 6, experiments are done with 8 seeds and both averaged learning curves and individual curves are plotted.

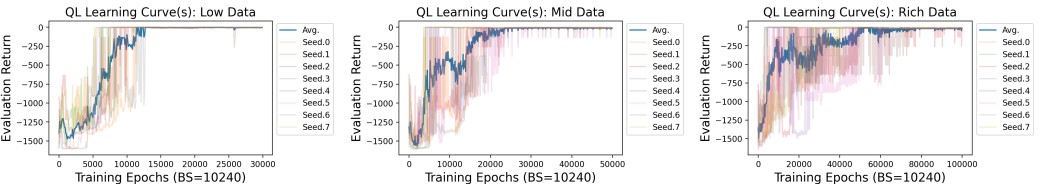

Figure 6: Q-Learning learning curves. When the size of offline data increases, more training epochs are needed for the convergence, and the stability of performance at convergence is reduced, leading to a larger variance and poorer performance in the rich-data regime.

## E   Additional Qualitative and Quantitative Results

### E.1   Adaptivity: Visualizing the Decisions of AOC and Quantitative Performance

We plot the control time trajectories in experiments of Section 5.4 in Figure 7. It is clearly shown that with a decreasing sampling rate of $\pi_1$'s trajectories (passing through the upper gate) in the offline dataset, the control time behaviors tend to choose more actions following the behaviors of $\pi_3$ (passing through the lower gate).

Quantitatively, Table 5 shows that while changing the sampling rate of trajectories from $\pi_1$, the success rate does not change much while the proportions of strategies chosen by the control policy vary accordingly.

Table 5: Quantitative results in the re-sampling and sub-sampling experiments. In all settings, 100 trajectories in total are generated using the proposed method. The success rate of reaching the goal and choices of solutions are presented in the table.

| Re/Sub-Sample | $\times 4$ | $\times 3$ | $\times 2$ | $\times 1$ | $\times 0.75$ | $\times 0.5$ | $\times 0.25$ |
|---|---|---|---|---|---|---|---|
| Success Rate | 0.91 | 0.97 | 0.90 | 0.92 | 0.92 | 0.93 | 0.92 |
| Passing Upper Gate | 76 | 69 | 61 | 42 | 31 | 24 | 17 |
| Passing Lower Gate | 15 | 28 | 29 | 50 | 61 | 69 | 75 |

### E.2   More Visualization Results and Extended Discussion on the Healthcare Dataset

We provide more qualitative results in Figure 8. In those figures, the colors yellow and blue indicate different treatments in the training data, separately.

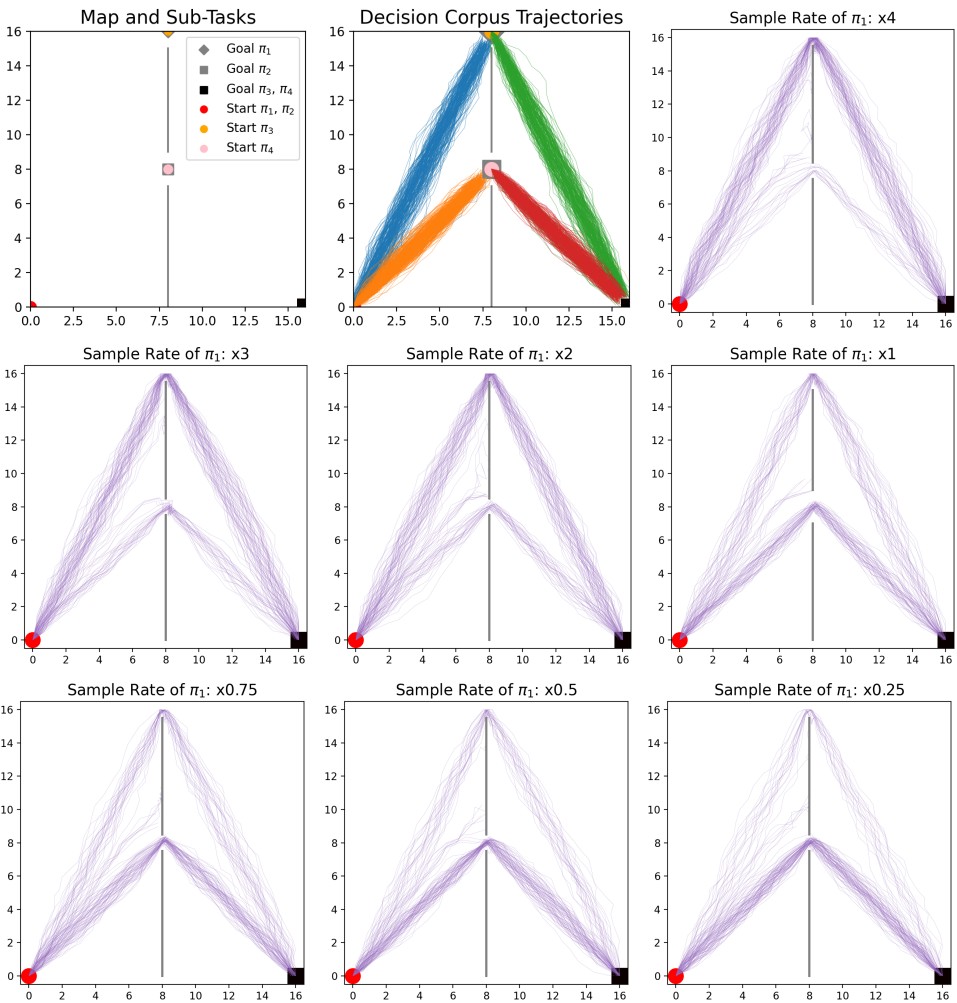

Figure 7: Visualization of control behaviors. The first two plots show the task and collected trajectories in the offline dataset. This is the same environment we have used in Section 5.3. The control time preferences can be controlled by changing the sub-sampling or re-sampling rate.

In the realm of healthcare, treatment decisions involving atypical patients — those with rare or non-standard characteristics that do not closely resemble the majority of training examples — present a significant challenge. These outliers often reside on the boundaries of existing treatment records of patients, rendering the optimal course of action ambiguous even for domain experts.

Our proposed method AOC provides a critical advantage in these complex scenarios. AOC constructs a minimal convex hull around such patients, unearthing potential risks associated with incorrect treatment approaches. In cases where the decision corpus spans disparate treatment groups, AOC reveals the impossibility of establishing a more representative convex hull that both contains the patient and exclusively includes members of the same treatment class.

Conversely, the k-nearest neighbors (kNN) approach fails to identify these high-risk patients. The nearest neighbors for such boundary cases can all belong to the same treatment group, thus offering no warning signals that the patient might be atypical and warrant special attention.

Therefore, AOC outperforms the kNN method not merely in terms of higher accuracy in suggesting treatments, but also in its ability to flag patients whose decision basis exhibits heterogeneity at the test time. In these instances, doctors or other human experts can give particular consideration, thereby enhancing the trustworthiness of the decision system.

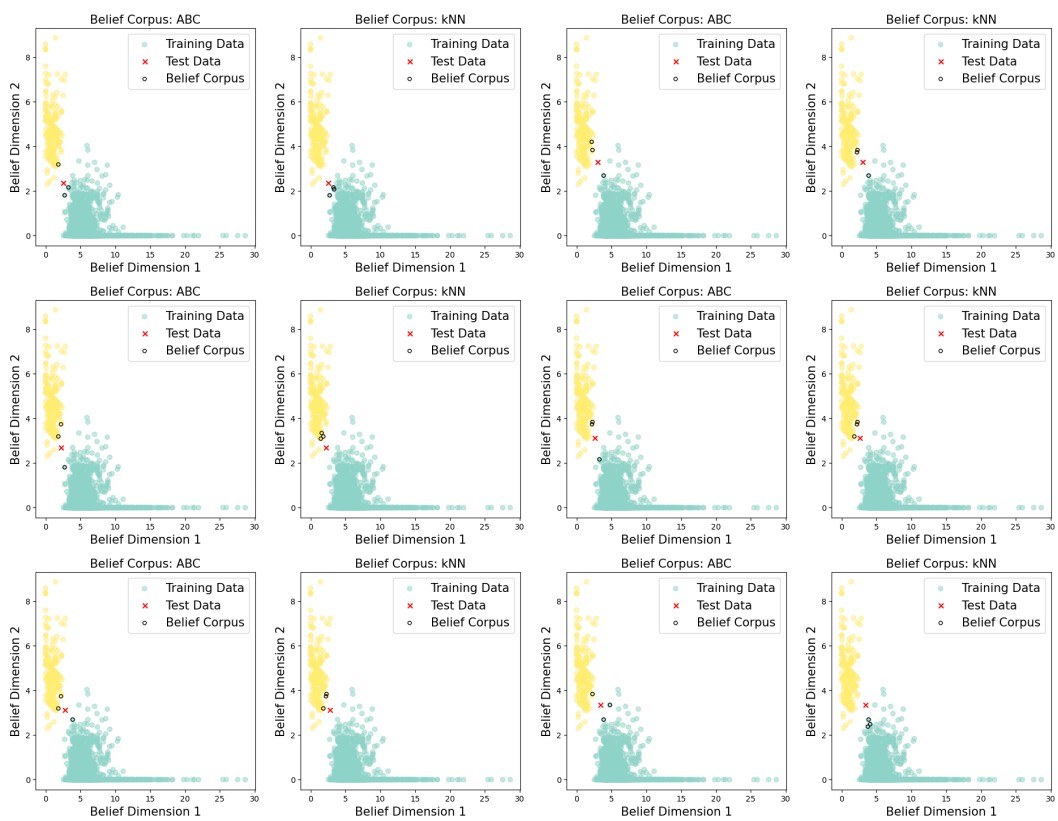

Figure 8: Visualization of the control time decision supports in the healthcare dataset. The two colors of the scatter plots denote different training time decisions. Test data is denoted by red cross marks, and the identified belief corpus subsets are marked with black circles.

## F  Additional Experiments

### F.1  Trade-Off between Accountability and Performance

**Experiment Setting**  In principle, the minimal convex hull in the $d$-dimensional belief space contains $d + 1$ examples. However, searching for such a minimal convex hull is a combinatorial optimization problem and can be extremely hard in high-dimensional space with many samples. In our implementation, we leverage a heuristic search method that constrains the combinatorial search inside the $k$-nearest neighbors of a given control time belief state.

We change such a hyper-parameter — the maximal number of examples in the corpus subsets can be used in building the minimal belief space convex hull. We experiment with $k = 1, 10, 20, 100$ separately.

**Results**  Table 6 shows the results. To make AOC work, the choice of $k$ should at least roughly match the dimension of the latent space. Otherwise, the aggressive extrapolation will hinder the performance of AOC, including using only the nearest neighbor as an approximation. While such a choice naturally trades off between explainability and decision quality, our empirical study shows using a redundant set of the corpus will hinder the performance — this demonstrates the necessity of using the minimal convex hull in AOC.

**Take-Away:**  *The minimal convex hull design in AOC is crucial for performance. We recommend the choice of size in constructing a minimal convex hull should match the dimension of the belief space.*

Table 6: The cardinality of the corpus subset is a hyper-parameter that trades off between accountability and performance. While in general using a larger number of corpus examples can better support the control decisions, it also has a risk of increasing the corpus residual in synthesizing the control time decision, hence hindering the performance of the control policy.

| $K$ | Performance |
|---|---|
| 1 | $-670.51 \pm 321.09$ |
| 10 | $-510.39 \pm 311.24$ |
| 20 | $-1.25 \pm 0.4$ |
| 100 | $-2.06 \pm 0.55$ |

## F.2   Visualizing Conservation of AOC

**Experiment Setting**   To better illustrate the conservative behaviors introduced by the hyper-parameter $\epsilon$, we conduct experiments on a Two Gates Maze environment to visualize the differences of using different choices of $\epsilon$'s.

In the Two Gates Maze task, an agent needs to navigate to a goal located at $(16, 8)$, the middle point of the right side wall, starting from $(0, 8)$, the middle point of the left side wall. Two gates open in a middle wall located at $x = 8$, hence the agent needs to pass one of those gates to reach the goal.

We generate an offline dataset from two behavior policies, each of which selects one of the two gates to pass the middle wall. The first two plots in Figure 9 show the map as well as the behavior trajectories as the dataset.

We then experiment with different choices of $\epsilon = [0.1, 0.3, 0.5, 0.7]$ in AOC to perform offline control.

**Results**   Results are shown in the last $4$ plots in Figure 9 (Purple lines denote the control time trajectories, ideal conservative policies' behaviors should be bounded by the behavior trajectories). In each setting, we roll out with 100 trajectories by AOC and visualize the behaviors, and report the averaged performance in Table 7. When a small $\epsilon$ is used, the control time behaviors show little extrapolation: trajectories are in general surrounded by the dataset behaviors. When $\epsilon$ becomes larger, more aggressive extrapolations emerge in control time behaviors, leading to poorer performances.

Table 7: The $\epsilon$ in Equation (11) contributes to the conservative behaviors of AOC.

| $\epsilon$ | Performance |
|---|---|
| 0.1 | $6.97 \pm 2.57$ |
| 0.3 | $2.58 \pm 5.13$ |
| 0.5 | $-3.10 \pm 1.00$ |
| 0.7 | $-3.20 \pm 0.00$ |

**Take-Away:**   *Using a small $\epsilon$ leads to more conservative behaviors in AOC. AOC performs offline control avoiding aggressive extrapolations by constructing the Minimal Hull and performing control with the decision corpora that have minimized residuals.*

## F.3   Additional Environment: LunarLanderContinuous

**Experiment Setting**   We additionally experiment on the LunarLanderContinuous-v2 environment [95] for the general interests of the RL community. The LunarLanderContinuous-v2 environment is a physics-based simulation game in which the agent must control a lunar lander to successfully land on a designated landing pad. It has a 2-dim action space and a 8-dim state space, both of which are continuous.

We compare AOC with the nearest-neighbor controller (**1NN**) [20] and its variant that using $k$ neighbors (**kNN**), model-based RL with Mode Predictive Control (**MPC**) [24], model-free RL (**MFRL**) [27], and behavior clone (**BC**) [25]. We change the size of the dataset to showcase the

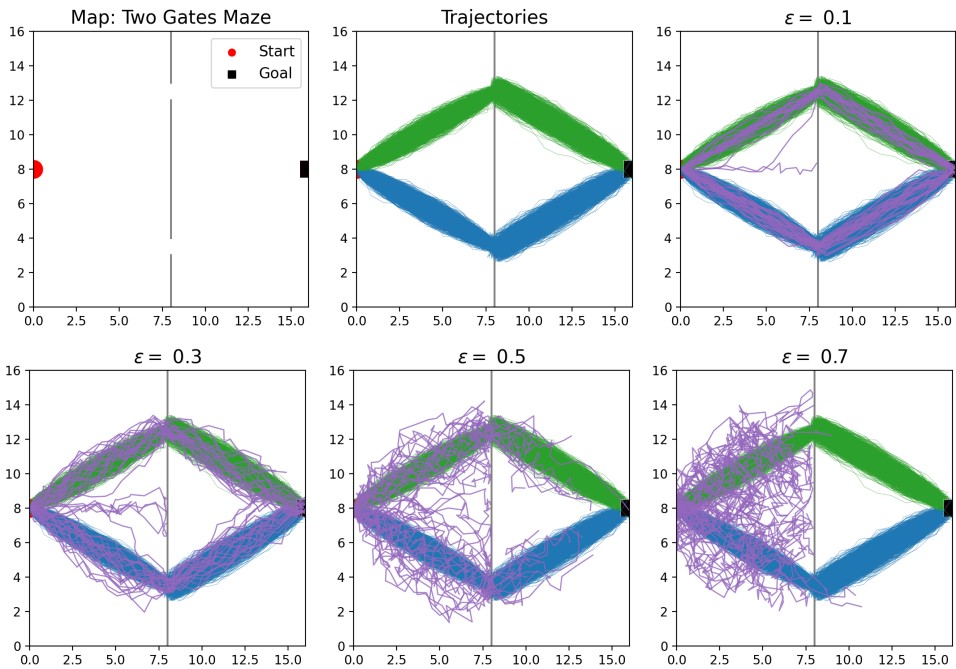

Figure 9: Visualizing the conservative behaviors controlled by the threshold controlled by a quantile number $\epsilon$. Using a smaller $\epsilon$ leads to more conservative behaviors hence benefiting the offline control problems where aggressive extrapolations can be dangerous.

performance difference of various methods under different settings. We experiment with different offline data availability, varying from 0.1M to 1M.

Table 8: Performance comparison on the LunarLanderContinuous-v2 environment under different settings (availability of offline data). The episodic cumulative reward of each method is reported. The last row (Data) reports the performance of trajectories in the offline dataset. **Higher is better.**

|        | 0.1M | 0.2M | 0.3M | 0.5M | 1M |
|--------|------|------|------|------|-----|
| AOC | $\mathbf{100.77 \pm 174.5}$ | $\mathbf{184.83 \pm 88.28}$ | $\mathbf{240.81 \pm 44.47}$ | $\mathbf{252.38 \pm 45.66}$ | $\mathbf{253.71 \pm 25.11}$ |
| kNN | $-86.07 \pm 172.57$ | $-43.87 \pm 69.1$ | $-3.04 \pm 116.76$ | $-59.63 \pm 143.33$ | $-31.47 \pm 152.72$ |
| 1NN | $-42.2 \pm 31.36$ | $-98.27 \pm 89.35$ | $-14.93 \pm 115.75$ | $-57.37 \pm 59.1$ | $-64.41 \pm 69.71$ |
| BC | $-130.23 \pm 33.23$ | $-118.61 \pm 42.18$ | $-36.91 \pm 57.32$ | $18.08 \pm 35.4$ | $54.46 \pm 69.16$ |
| MFRL | $\mathbf{92.0 \pm 82.05}$ | $\mathbf{165.11 \pm 54.48}$ | $\mathbf{221.3 \pm 19.63}$ | $219.55 \pm 38.97$ | $254.55 \pm 24.42$ |
| MPC | $-31.42 \pm 32.09$ | $-35.52 \pm 43.09$ | $-50.96 \pm 22.96$ | $-67.88 \pm 55.49$ | $-96.22 \pm 85.01$ |
| Data | $149.99 \pm 133.45$ | $207.92 \pm 58.34$ | $171.19 \pm 100.43$ | $169.42 \pm 96.02$ | $184.92 \pm 69.59$ |

**Results** We present the results in Table 8. In all settings, we find AOC is able to achieve an on-par performance of black-box decision-making algorithms, and outperforms the accountable baselines. In this environment, we find the model-based approach is not able to learn a well-performing reward function approximator. On the other hand, different from the Pendulum-Het settings where the stability of the balanced state is essential in achieving high performance, the stability of the model-free method in this environment is not an issue.

**Take-Away:** *The results on the LunarLanderContinuous environment again demonstrate the desired properties of AOC: while being accountable, it achieves similar performance as the black-box learning algorithms and is robust under different data availability.*

### F.4 Black-Box Policy as More Efficient Sampler

**Experiment Setting**    In the main text, we introduce the uniform sampling approach for control time execution. While such an approach is simple and effective in our accountability-sensitive control benchmark tasks, it can be inefficient in high-dimensional continuous control tasks. For the interest of the general continuous control community, we experiment with a higher dimensional task in this section to stress test the capability of AOC in more challenging tasks.

We experiment on the BipedalWalker-v3 environment that has 24-dim observational space and 4-dimensional action space. Different from previous experiments where a uniform sampler is applied, in this section, we use black-box models as the more efficient sampler and leverage AOC as a post-hoc interpreter for decision accountability. Specifically, we compare AOC with a uniform sampler (**AOC-Uniform**) and AOC with Behavior Clone policy as a black-box sampler (**AOC-BC**) against the same baselines as previously, i.e., the nearest-neighbor controller (**1NN**) [20] and its variant that using $k$ neighbors (**kNN**), model-based RL with Mode Predictive Control (**MPC**) [24], model-free RL (**MFRL**) [27], and behavior clone (**BC**) [25]. To improve the black-box controller's performance and hence isolate the source of gain, we use the offline dataset of size 1M.

**Results**    Results are reported in Table 9. In this high-dimensional control task, the uniform sampler is inefficient and fails to converge to a well-performing policy in control time. Among all black-box methods, BC achieves the best performance. And AOC with BC as its sampler achieves improved performance, while at the same time being accountable.

Table 9: Performance on the BipedalWalker-v3 environment. The episodic cumulative reward of each method is reported. **Higher is better.**

| Method | Performance |
|---|---|
| kNN | $-109.72 \pm 5.85$ |
| 1NN | $-111.95 \pm 8.25$ |
| AOC-Uniform | $-90.44 \pm 14.06$ |
| AOC-BC | $\mathbf{276.98 \pm 82.78}$ |
| BC | $\mathbf{208.72 \pm 95.72}$ |
| MFRL | $18.51 \pm 111.53$ |
| MPC | $-96.82 \pm 24.7$ |
| Data-Avg-Return | $202.25 \pm 102.61$ |

**Take-Away:**    *AOC can work both in isolation or combined with black-box policies. AOC can be used as a plug-in to add accountability to black-box controllers in a post-hoc manner. In high-dimensional control tasks, uniform sampling can be inefficient and black-box samplers can alleviate such a difficulty.*

### F.5 Identify Control Time OOD Examples with AOC

**Experiment Setting**    In this section, we show that AOC can be applied to OOD example identification [96] during control time. Specifically, we conduct the experiment with the **BipedalWalker** environment. During control time, a black-box controller described in the last section is used, and we start to inject Gaussian noise into the control actions as disturbance after the 500-th timestep to create an OOD scenario.

**Results**    We repeat the above process for 10 times and report the averaged step-wise instant reward curve and corpus residual curve during rollouts in Figure 10. We note that in the figure in the beginning steps, the walker starts from a static state to walk, thus leading to an increase in the instant reward curve. We then label the control steps between the 100-th and the 500-th step as the stable walking phase, during which the walker receives a nearly constant instant reward (around 0.4 per step), and the corpus residual during this period is stable and close to 0. The Gaussian noise is injected after the 500-th timestep, leading to a clear increase in the corpus residual curve and a sudden decrease in the instant reward curve. According to the corpus residual values' sudden increase, the

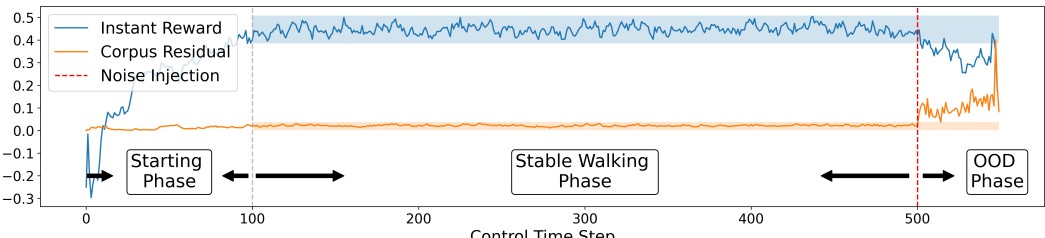

Figure 10: AOC can be used for OOD detection in control time. The corpus residual values can be used to detect large epistemic uncertainty on OOD examples. Shaded areas in the figure denote the **3-sigma interval**.

control time OOD examples can be identified according to the *3-sigma rule of thumb*. (The 3-sigma intervals are marked with shaded areas in the figure).

**Take-Away:** *Control time OOD examples can be identified by AOC according to the sudden increases in the corpus residual value.*

## Limitations and Future Work

In this study, our exploration of accountable offline control primarily targets low-dimensional control tasks, drawing inspiration from healthcare applications where treatments are typically of low dimension. However, the uniform sampling approach we propose may not perform as well in high-dimensional control systems, which can be of interest in the field of robotics study. We do offer an initial exploration of using black-box samplers in AOC within our appendix, but there remains ample room for further work in enhancing the efficiency of the sampling process. Additionally, there's potential for expanding AOC into online control settings by combining it with optimism in the face of uncertainty (OFU) explorers, to pursue accountable online control. This, however, lies beyond the scope of our current paper.

## Broader Impact

In this study, we examine the offline control problem, which holds significant potential for applications in costly, safety-sensitive, and critical domains such as healthcare and finance. While previous works have primarily focused on efficient learning in offline settings, the accountability of offline decisions remains largely unexplored despite its importance.

In critical domains like healthcare, it's vital that decisions are based on supportive evidence. For instance, when a patient is treated in a certain manner, it should be based on the successful outcomes of previous patients with comparable conditions who received the same treatment. The ability to trace the supportive basis of decisions enhances the process of policy reasoning and debugging, thereby improving the trustworthiness of decision-making systems.

However, we bring to light the potential risk associated with applying AOC to critical real-world decision-making systems. This risk stems from potential heterogeneous outcomes, i.e., the aleatoric uncertainty associated with decision outcomes. For example, similar patients undergoing the same treatment may experience different results. Therefore, when the variance of outcomes in the corpus subset is high, users should exercise caution regarding the potential heterogeneous outcomes of decisions.

