# OpenReview forum: "Accountability in Offline Reinforcement Learning: Explaining Decisions with a Corpus of Examples"
_NeurIPS.cc/2023/Conference — NeurIPS 2023 poster_

### Official Review · Reviewer_qevo · 2023-07-06

**Soundness:** 3 good
**Presentation:** 2 fair
**Contribution:** 2 fair
**Rating:** 4
**Confidence:** 3

**Summary:**

This paper poses the problem of learning a controller from a batched dataset containing time-series observations of the world, actions, and a value estimate (e.g., for determining how to treat a patient given the results of medical tests based upon patient outcomes). This problem is considered with a POMDP formalism with execution traces. The paper imposes assumptions about the linearity of the mapping from the belief space to the value space in order to create some analytical properties about how wrong the model might be or to mitigate the model error. The algorithm, ABC, is evaluated against baselines in a set of batched-data experiments, including in a healthcare applicaiton.

**Strengths:**

+The paper addresses an important problem of bringing safety to machine learning

+The paper clearly states assumptions and presents logical arguments and definitions to support its thesis.

+The evaluation shows positive results and does so in important domains (e.g., healthcare)

**Weaknesses:**

-The paper seems to be addressing the problem of offline reinforcement learning without actually addressing offline reinforcement learning. Though, this paper does cite a plethora of papers that address this topic. As such, it is difficult for the reviewer to properly contextualize this paper in this relevant prior work.

-The paper appears to be missing a number of baselines for offline reinforcement learning, as shown below. Some of the baselines chosen do not have access to the same set of information available to the ABC algorithm (e.g., the BC model).

Chen, L., Lu, K., Rajeswaran, A., Lee, K., Grover, A., Laskin, M., Abbeel, P., Srinivas, A. and Mordatch, I., 2021. Decision transformer: Reinforcement learning via sequence modeling. Advances in neural information processing systems, 34, pp.15084-15097.

Kumar, A., Zhou, A., Tucker, G. and Levine, S., 2020. Conservative q-learning for offline reinforcement learning. Advances in Neural Information Processing Systems, 33, pp.1179-1191.

-The paper develops theory to bring accountability to this problem. However, the results section provides some relatively simple computational examples and a qualitative description that is in the eye of the beholder. It would have been better to provide a clearer, more convincing test to demonstration that there are clear guarantees and fulfilled analytical properties.

-One could have considered a human-subject experiment to evaluate whether this approach is really "accountable." Literature on accountability could have been considered as well. For example, see:

Kim, B. and Doshi-Velez, F., 2021. Machine learning techniques for accountability. AI Magazine, 42(1), pp.47-52.

**Questions:**

-Why is this problem different than offline reinforcement learning?
-Why are offline RL baselines not included?
-How do the results provide convincing evidence of accountability in a non-superficial manner?
-For what class of problems is the linear assumption for the mapping from belief to value reasonable?
-What is the computational complexity of the approach?
-How scalable is the approach with the size of the state space?
-By relying on belief space modeling, why is this approach relevant to large, real-world problems?

**Limitations:**

The paper does not mention "limit" once.

---

> ### Author Rebuttal · Authors · 2023-08-08
>
> We thank the reviewer for the time and effort in reviewing our paper. We will respond to each question in turn:
>
> ---
> ### Q1: Why is this problem different than offline reinforcement learning?
> A1: In this paper, we study the accountability of offline decision-making in high-stake systems, and applied the proposed method to real-world healthcare dataset. ABC performs a form of offline reinforcement learning, but unlike existing work on offline RL, it seeks to achieve P1-P5 below.
> As we have emphasized in our abstract, introduction, related work, and experiment section that what makes ABC different from existing literature including Offline-RL is its 5 properties:
> - P1: controllable conservation to avoid aggressive extrapolation;
> - P2: accountability that provides a decision basis;
> - P3: suitability for low-data regimes;
> - P4: adaptability to user specifications that allows customization;
> - P5: flexibility in strictly batched imitation settings for broader applicability.
>
> Out of those 5 properties, **Offline-RL only satisfies P1**.
>
> We also discussed the similarities and differences of ABC to Offline-RL in the section of Extended Related Work (Appendix B).
>
> ### Q2: Why are offline RL baselines not included?
> A2: The only experiment in which Offline-RL could be compared with ABC is Sec.5.1. In all other experiments, Offline-RL does not enjoy the properties we discussed in the paper. Nor do those Offline-RL algorithms address the issue of accountability in decision-making.
>
> We **conducted additional experiments** to better address the reviewer's concern and provide **results in the attached PDF file** due to space limitations.
>
> ### Q3: How do the results provide convincing evidence of accountability in a non-superficial manner?
>
> A3 :  (1) In our main text, we use Sec.5.3 to highlight the accountability of the proposed method. Where we can have the conclusion that ABC's decision-making process exhibits strong accountability, as the **corpus subset can be tracked at every decision-making step**. This is evidenced by ABC's successful completion of a multi-stage Maze task, wherein it effectively learns from mixed trajectories generated by multiple policies at differing stages.
>
>  (2) Apart from the above qualitative results that permit visualization, we also provided additional empirical evidence on the property of accountability in Section 5.5, where we study the real-world healthcare application of ABC, and highlight how **accountability helps to identify boundary examples in decision-making**.
>
>  (3) Moreover, in Appendix F.5, we demonstrate how to leverage the **accountability of ABC to identify OOD examples**, this is another evidence showing the accountability of our proposed method.
>
> ### Q4: For what class of problems is the linear assumption for the mapping from belief to value reasonable?
>
> A4: **Our Remark 3.3 answers this question.**
>
> The linear relationship between the latent belief space and the value is a property rather than an assumption. As we’ve stated in Remark 3.3, this property is often the case with neural network approximators, where the belief state is the last activated layer before the final linear output layer.
>
> ### Q5: What is the computational complexity of the approach?
>
> A5: **Our discussion in Appendix D.3 and D.5 can answer this question.**
>
> Due to the page limit, we postponed the discussion of computational complexity, hardware requirement, and wall-clock running time in Appendix D.3 and D.5, respectively.
> With our proposed solution, the convex hull decomposition takes less than 10 seconds with a uniform sampler that samples 100 actions randomly for every time step. Increasing the number of sampled actions will lead to a sub-linear increase in computational time with parallelization.
>
> ### Q6: How scalable is the approach with the size of the state space?
> A6: **Our experiments in Appendix F.3-F.5 are designed to answer this question.**
>
> **ABC can work both in isolation or combined with black-box policies.** ABC can be used as a plug-in to add accountability to black-box controllers in a post-hoc manner. In high-dimensional control tasks, uniform sampling can be inefficient and black-box samplers can alleviate such a difficulty.
>
> ### Q7: By relying on belief space modeling, why is this approach relevant to large, real-world problems?
> A7: In this study, we examine the batch control problem, which holds **significant potential for applications in costly, risk-sensitive domains such as healthcare and finance.** While previous works have primarily focused on efficient learning in batch settings, the accountability of offline decisions remains largely unexplored despite its importance.
>
> In healthcare, it's vital that decisions are based on a supportive basis. For instance, **when a patient is treated in a certain manner, it should be based on the successful outcomes of previous patients with comparable conditions who received the same treatment.** The ability to trace the supportive basis of decisions enhances the process of policy reasoning and debugging, thereby improving the trustworthiness of decision-making systems.
>
> While technically ABC is built on top of belief space modeling, the accountability generated by ABC is instance-level based on the one-to-one mapping between examples and belief states. Hence, **the use of belief space modeling in ABC does not restrict its applicability or effectiveness in real-world problems.** Therefore, the modeling of belief space enables a more nuanced understanding of individual decisions, aligning with the complexity of real-world scenarios.
>
> ### Q8: The paper does not mention "limit" once.
> A8:  In fact, we do have a discussion section **Appendix G. Limitations and Future Work** on Page 27 of our paper.  We have updated our conclusion and refer readers to Appendix. G for limitations.
>
> ---
> We hope that these clarifications address your concerns, and we are happy to have further discussions would they remain unclear.

---

> > ### Comment · Reviewer_qevo · 2023-08-14
> > **Missing attachment**
> >
> > Unless I am mistaken, I believe the authors were going to attach a document: "We conducted additional experiments to better address the reviewer's concern and provide results in the attached PDF file due to space limitations."
> >
> > However, I don't believe I see this document. Can the authors confirm?

---

> > > ### Author Response · Authors · 2023-08-14
> > > **Response to Reviewer qevo**
> > >
> > > Dear Reviewer qevo,
> > >
> > > We sincerely thank you for your follow-up response.
> > >
> > > According to the rebuttal guideline, we provided the one-page pdf attachment in the **overall response**. Please see the **Author Rebuttal by Authors** at the very top of this page, and the attachment we mentioned can be downloaded there.
> > >
> > > Many thanks for your consideration. Please let us know should there be any additional questions or concerns, we are more than willing to provide further explanations.
> > >
> > > Regards,
> > > Authors

---

> > > > ### Comment · Reviewer_qevo · 2023-08-17
> > > > **Thanks**
> > > >
> > > > I thank the reviewers to pointing me to this additional material. Based upon these responses as well as the responses for author co-authors, I will initially raise my score and wait for further discussion with the other reviewers.

---

> > > > > ### Author Response · Authors · 2023-08-17
> > > > > **Dear Reviewer qevo**
> > > > >
> > > > >
> > > > > We deeply appreciate the reviewer's continued engagement and consideration given to re-evaluating our paper, and the _initial_ raising of the score. We have made sincere efforts to address the concerns in our previous response. That said, it seems that there may still be some reservations on the reviewer's part.
> > > > >
> > > > > Recognizing that we still have a few days left in the discussion phase, might we kindly request the reviewer to pinpoint any outstanding concerns or specific questions (in particular which reasons to reject still outweigh the ones to accept since “4: Borderline reject” means
> > > > > _"Technically solid paper where reasons to reject, e.g., limited evaluation, outweigh reasons to accept, e.g., good evaluation. Please use sparingly."_) ?
> > > > >
> > > > > We would address these with the utmost diligence in the hope that the reviewer will consider raising the score further since we believe the paper goes beyond a
> > > > > borderline reject.

---

### Official Review · Reviewer_fQBY · 2023-07-06

**Soundness:** 2 fair
**Presentation:** 2 fair
**Contribution:** 2 fair
**Rating:** 5
**Confidence:** 3

**Summary:**

This paper investigates imitation learning in scenarios with limited data. The proposed approach (ABC) involves utilizing a linear combination of the belief space to generate accountable decisions.
The authors evaluate the performance of ABC in simulated and real-world healthcare scenarios, highlighting its ability to effectively handle batched control tasks while maintaining a high level of performance and accountability.

**Strengths:**

1. The paper is presented in a clear and easily understandable manner.
2. This article investigates a highly meaningful direction, which generates reliable strategies through offline data.

**Weaknesses:**

1. Assuming that the value is a linear combination of belief states is a strong assumption, which may not hold true in certain tasks, particularly those that are harder and more complex.
2. The optimization problem (Eq. 9) seems to be ill-defined. Eq. 9 could have an infinite number of solutions because the effect of $l\circ b$ is equivalent to $cl \circ c^{-1}b$, where $c$ represents a scalar. This would have an impact on the distance calculation when solving Eq. 10. It would be beneficial to address the limitations of the optimization procedure and conduct additional experiments to examine the influence of different solutions to the optimization problem (Eq. 9).
3. The paper lacks a discussion and comparison with offline reinforcement learning, which is an important related work and a strong baseline.
4. The evaluation is confined to simple environments. For more challenging tasks, please refer to [1].

Reference:
[1] Fu, Justin, et al. "D4rl: Datasets for deep data-driven reinforcement learning." arXiv preprint arXiv:2004.07219 (2020).

**Questions:**

1. Could you provide the results of offline reinforcement learning methods in the test environments?

2. Can you provide the environment results obtained when applying the proposed approach to more challenging environments?


**Limitations:**

This work is primarily focused on simple environments and does not address more complex scenarios.

---

> ### Author Rebuttal · Authors · 2023-08-08
>
> We thank the reviewer for the time and effort in reviewing our paper. We will respond to each point in turn:
>
> ---
> ### 1. Property 3.2 is not an Assumption
> Property 3.2 is **NOT an assumption**, and is **INDEPENDENT TO SPECIFIC TASKS**. It is about the architecture of the neural network being used (not an assumption imposed on the environment).
>
> We wish to clarify that the belief state in our context is the **values of the last activation layer in neural networks**, hence it is independent of the task. As long as there is a neural network with a linear layer as the output layer that approximates the value function, we are able to use its last activation layer as our belief state.
>
> Therefore, we would like to note that our property 3.2 is not an assumption imposed on the environment, but a property that is satisfied by general neural network approximators, as we have noted in Remark 3.3.
>
> ### 2. Eqn. (9) is Well-Defined.
>
> Eqn. (9) depicts the MSE minimization using neural networks. It’s a well-defined problem.
>
> To be specific, we consider a neural network with a linear output layer. We denote the layers before the last as $\mathbf{b}$, and the linear output layer as $\mathbf{l}$. We explicitly write the value function approximation minimization process in Eqn.(9). We would like to note that, different solutions of this equation correspond to different neural network parameters. And any set of parameters that minimized the loss function can be used in practice.
>
> We do not assume the uniqueness of the optimized neural network parameters, and our optimization process following Eqn.(9) will not be affected.
> To see this, suppose we have $\mathbf{b}’, \mathbf{l}’ = c^{-1}\mathbf{b}, c\mathbf{l}$, Eqn.(10) and Eqn.(11) will still get the identical results. This is because the calculations following Eqn.(9) only use the relative information among beliefs, rather than their absolute values.
>
>
> ### 3. We added Offline RL Algorithms as Additional Baselines
>
> **The only experiment in which Offline-RL algorithms can be compared with ABC is Section 5.1.** In all other experiments, Offline-RL **does not enjoy the properties we discussed in the paper.** Nor do those Offline-RL algorithms address the issue of accountability in decision-making.
>
> To address the reviewer's concern, we additionally compare ABC with CQL and TD3-BC in Section 5.1. Not surprisingly, we find the performance of the original CQL and TD3-BC on Pendulum-Het are poor. This is because they are not designed for partial observable tasks with high stochasticity.
> Therefore, we further implemented an improved version of the TD3-BC with a recurrent context encoding module [cf. Meta-Q-Learning]. In those experiments, we find the learning process of Offline-RL is not stable, leading to a large variance in the policy quality — we have observed a similar problem in our previous MFRL baseline, and reported it with details in Appendix D.7.
>
> _The cumulative reward of each method is reported. **Higher is better.**_
>
> | Task              | Low-Data            | Mid-Data            | Rich-Data           |
> |-------------------|---------------------|---------------------|---------------------|
> | ABC               | **-1.39 ± 1.39**    | **-1.25 ± 0.40**    | **-0.6 ± 0.08**     |
> | BC                | -422.77 ± 409.51    | -225.32 ± 340.83    | -126.74 ± 280.73    |
> | TD3               | -4.1 ± 2.76         | -11.95 ± 4.68       | -15.27 ± 6.46       |
> | CQL               | -793.89 ± 206.0     | -889.85 ± 291.99    | -805.49 ± 578.75    |
> | TD3-BC            | -844.95 ± 170.93    | -781.8 ± 337.11     | -821.02 ± 587.77    |
> | TD3-BC-Recurrent  | -82.92 ± 115.79     | -41.92 ± 59.28      | **-0.45 ± 0.57**    |
> | Data-Avg-Return   | -307.81 ± 387.53    | -245.54 ± 338.65    | -208.81 ± 272.84    |
>
> ### 4. Difference between ABC and Offline-RL
>
> We would like to note that Offline-RL is related to our work (Appdx. B.1), yet it is not our primary focus. The mentioned D4RL benchmark is famous in Offline-RL, yet it presents a weak link to the main problem of accountability studied in this work. In fact, **the only experiment that Offline-RL algorithms can be compared with ABC is Section 5.1.** In all other experiments, Offline-RL does not enjoy the properties we discussed in the paper.
>
> We wish to clarify that **we mainly studied the accountability of offline decision-making in high-stake systems, and applied the proposed method to real-world healthcare dataset.** This is different from the normal pursuit of Offline-RL that aims at improving conservative value estimation.
>
> To make it more explicit, we emphasized in our abstract, introduction, related work, and experiment section that what makes ABC different from existing literature including Offline-RL is its 5 properties:
> - P1: controllable conservation to avoid aggressive extrapolation;
> - P2: accountability that provides a decision basis;
> - P3: suitability for low-data regimes;
> - P4: adaptability to user specifications that allows customization;
> - P5: flexibility in strictly batched imitation settings for broader applicability.
>
> Out of those 5 properties, Offline-RL only explicitly satisfies P1.
>
> To address the reviewer's concern about more challenging tasks, we **refer to additional experiments in Appendix F.3, F.4, F.5**, where we demonstrate the scalability of ABC and the potential of combining ABC with black-box policies. From such a perspective, ABC can be used as a post-hoc interpretation mechanism for any given black-box algorithm like CQL, TD3-BC, etc. This could be interesting and important future work but is out of the scope of this methodology paper focusing on accountability, rather than Offline-RL.
>
> ---
> We hope that these clarifications address your concerns, and we are happy to have further discussions would they remain unclear.

---

> ### Author Response · Authors · 2023-08-15
> **Further Discussions and Feedback Welcome**
>
> We deeply appreciate the insights you've shared during the review process. Following our revisions and previous responses, we are genuinely curious if we have adequately addressed the concerns you raised.
>
> Should there be any leftover questions, concerns, or areas you feel need more clarification, please do not hesitate to let us know. We greatly respect your insights and stand ready to make any additional refinements based on your feedback.

---

> ### Author Response · Authors · 2023-08-16
> **Gentle Reminder: Feedback on Our Submission**
>
> It's been over a week since we shared our response addressing each point from the prior review comments, and we haven't yet had the privilege of receiving your feedback. To ensure we maximize the discussion period, we gently reach out, hoping the reviewer might engage further. We would appreciate it a lot if the reviewer could please let us know if there are any additional concerns or areas that need our clarification.
>
> ---
> In order to assist the reviewer in recalling the specifics of the work, the previous comments, and our response, we would like to offer a brief summary of each aspect:
>
>  ## Summary of Our Work
>
> In our work, we study the problem of accountability in batched control tasks. We proposed the Accountable Batched Controller (ABC), which goes beyond previous literature studying offline-RL by having 5 unique properties that are desired in responsibility-sensitive scenarios:
> - P1. Conservation;
> - P2. Is accountable
> - P3. Works with Low-Data;
> - P4. Being adaptive to user specification;
> - P5. Works in strictly batched imitation setting;
>
> We demonstrated those properties through extensive empirical studies: we verify each of those properties through separate experiments, each of which at least contains two environments with varying set-ups. We highlight the use case of ABC in a real-world healthcare dataset.
> Additionally, we provide more qualitative results in the appendix, and extension of ABC to work as a post-hoc plug-in for understanding black-box policies.
>
>
> ## Summary of Comments and Our Responses
>
> In your previous comments, you mainly had the following questions, and we answered those questions through the previous response:
>
> ### Point 1
> You thought the linear relationship between the belief and value is an assumption, **unfortunately, by mistake.**
>
> ### - Our Response
> We have pointed out that **this is a misinterpretation**. We **DO NOT** impose any linear assumption on the tasks, and our linear decomposition is independent of the complexity of tasks. Property 3.2 and remark 3.3 highlighted the general applicability of our proposed method: the existence of such linear decomposition is a property of neural value estimators as long as they apply a linear output layer, which is the most common implementation in the field.
>
> ### Point 2
> You thought the Eq.(9) is not well-defined, **unfortunately, by mistake.**
>
> ### - Our Response
> We have argued that **this is not true** as Eq.(9) depicts the MSE minimization objective. Please kindly let us reiterate that we do not impose uniqueness constraints on optimizing neural networks (as no one would do so). Any optimizer optimizes Eqn. (9) is enough for our algorithms to work.
>
>
> ### Point 3
> You commented we should make a comparison with more offline-RL baselines. **We have added experiments as suggested.**
>
> ### - Our Response
>  We first argued that there are clear differences between the general interests of offline-RL and our work: **ABC pursues the 5 unique properties** desired for responsibility-sensitive decision-making systems, offline-RL mainly focuses on **1 of those 5** aspects — the conservative learning. We acknowledge the importance of research in offline-RL and their challenging environments, yet we believe **both of those topics are important and warrant their individual study to make scientific progress.**
>
>
> In order to better address your concerns, we have **followed your suggestion to provide additional experiments** by comparing CQL and TD3-BC, two of the most prevailing offline-RL algorithms, as additional baselines in section 5.1. Kindly allow us to emphasize that this section is the only section that offline-RL can be compared to our method. **For all other sections (5.2-5.6), the characteristics we have underscored are unique to our method and not inherent to offline-RL algorithms, making them non-comparable.**
>
>
> In our additional empirical findings, we demonstrated that **conventional offline-RL algorithms do not serve as well-performing baselines** for the problems we studied in this work. Consequently, we further improved the TD3-BC, enhancing its ability for tasks that necessitate **historical transition memory and competence in addressing stochasticity.** These attributes are paramount, especially in the domains we've focused on, such as healthcare. Our results indicate that offline-RL algorithms tend to prefer larger datasets, as data scarcity could severely impact stability. We would release our advanced TD3-BC implementation with the community, offering yet another valuable resource.
>
>
>
> ---
> We genuinely value your perspective and were wondering if there might be any outstanding questions or concerns we can assist with. Your insights are of utmost importance to us, and we aim to make the most of the available time to address any points you may raise.

---

> ### Comment · Reviewer_fQBY · 2023-08-18
>
> I am deeply appreciative of the author for their response, which effectively addresses several of my concerns.
>
> I have made corresponding adjustments to my score (3 to 5).
>
> However, I really suggest the authors reorganize the presentation of the paper since it is quite confusing to me.

---

> > ### Author Response · Authors · 2023-08-18
> > **Thanks for Sharing Further Suggestions**
> >
> > We sincerely thank the reviewer for the encouraging feedback and kind consideration in re-evaluating our work.
> >
> > In response to your suggestions regarding the presentation and to enhance the clarity of the paper, we have made a series of updates to our manuscript. We have detailed these changes in the official comment titled "Follow-Up Author Response on Presentation to Reviewer fhj3 and fQBY".
> >
> > Our revision mainly includes 1. reorganization of the introduction; 2. a method sketch paragraph before introducing the method; and 3. extended related work that focuses on distinguishing ABC from offline-RL literature. We genuinely hope our revised manuscript could meet your expectations and provide clarity for all readers.
> >
> > We would appreciate it if you could kindly let us know if there were any further concerns or suggestions on the presentation. In the limited time remaining, we are still eager to do our utmost to address them!
> >
> > Regards,
> >
> > Authors

---

### Official Review · Reviewer_izcS · 2023-07-07

**Soundness:** 3 good
**Presentation:** 2 fair
**Contribution:** 3 good
**Rating:** 6
**Confidence:** 3

**Summary:**

This work proposes accountable batched control with five desirable properties. The design is motivated by the fact that the reward or feedback of trajectories are hard to obtain in high-stake responsibility-sensitive applications. The minimal hull subset of the decision corpus is constructed for the decomposition of the value function for each candidate action. Then the optimal policy is selected in terms of the weighted value function. This work demonstrates the promise of the five properties on one real-world healthcare dataset and one simulated maze environment.

**Strengths:**

After the rebuttal I tentatively raised my score from 5 to 6.

---

The paper is quite obscure and not easy to understand, making it challenging to grasp the complete understanding of the methodology it presented. However, I would appreciate it that the authors included a video presentation in the appendix and saved my time. In particular, the animation of the algorithm makes it easier to understand.

The introduction of the batched control is a seemingly novel contribution. The experimental setup and analysis are presented in a concrete and well-written manner, albeit deferred to the appendix due to space constraints. In addition, the authors attached an anonymous link to their code implementation.

**Weaknesses:**

While I may not have a comprehensive understanding of the literature, it appears that the focus of this paper leans more towards reinforcement learning rather than the chosen primary area of interpretability and explainability.

I am uncertain about the reasons behind the batched controller possessing the five advantageous properties in comparison to other well-known methods, including Q-learning, model-based RL, and behavior clone. Although I skimmed through the Appendix B, the interpretation is still unclear.

**Questions:**

1. The term *decision corpus* is never explained. How does it differ from the trajectory in the offline data?
2. The batched controller appears to neglect the historical decisions and fails to account for the temporal correlation among actions. How can you guarantee that the policy will receive the optimal accumulated rewards?

**Limitations:**

The authors have include a separate broader impact section in the Appendix.

---

> ### Author Rebuttal · Authors · 2023-08-08
>
> We thank the reviewer for the time and effort in reviewing our paper. We will respond to each point in turn:
>
> ---
> ### 1. Definition of the Decision Corpus
> The definition of _Decision Corpus_ is explained in line 6 in the abstract and line 61 in Sec. 3. In our context, we use _Decision Corpus_ to refer to the offline decision dataset.
> We have improved the clarity of the definition and made it more explicit in our revision.
>
> ### 2. Historical Decisions are Captured by the Belief State
> - **[No temporal correlation among actions]** We would first note our work is developed under the general Partially Observable Markov Decision Process (POMDP) setting, denoted as a tuple $(\mathcal{X}, \omega, \mathcal{O}, \mathcal{A}, \mathcal{T}, \mathcal{R}, \gamma, \rho_0)$. Under the Markovian property, **there should be no temporal correlation among actions.**
> - **[Ability to capture historical decisions]** In such a setting, it is important to capture the historical decisions and transitions, therefore, we introduced the observational transition history variable $h_t$, defined as $h_t = (o_{<t}, a_{<t}, r_{<t}) \in \mathcal{H} \subseteq \mathbb{R}^{(d_o+d_a +1)\cdot (t-1)}$. In our work, **the belief mapping $\mathbf{b}$ is a function of observation, action, and historical transition**, i.e., $\mathbf{b} = \mathbf{b}(o_t, a_t, h_t)$, to capture the information in historical observations, decisions and return.
> - **[Optimize accumulated rewards]** In order to optimize the correct learning objective of maximizing cumulative episodic return, **we use the cumulative return as $v^c$ in Eqn.(2).**
> As a consequence, the estimated value $\hat{v}$ approximates the true cumulative return, and ABC optimizes this cumulative reward proxy in decision-making.
>
> ### 3. Extended Discussion Comparing ABC with Existing Algorithms
> Below, we discuss each of the desired properties in turn. For each of the properties, we start with introducing the definitions of the property, followed by comparisons among ABC and MFRL (Q-learning), MBRL, and BC.
>
> (1) **Accountability**: the decision-making process is traceable, and the decisions can be supported by concrete examples in the offline dataset.
>   - ABC: the decisions of ABC are accountable, since they are generated referring to the Corpus Subset within the minimal convex hull. Those examples provide an example-based explanation of the decisions.
>   - MFRL: In MFRL, a black-box value network and black-box policy network are learned with the offline dataset. There is no decision support for the black-box policies.
>   - MBRL: In MBRL, a black-box world model optimized with the offline dataset is used as a proxy of the true dynamics, and planning algorithms are then applied to such a black-box model to make decisions. Those decisions are not supported by explicit references.
>   - BC: In BC, a black-box policy is learned through supervised learning. The output of such a policy is hard to be linked with specific training examples.
>
> (2) **Conservation**: estimations of decision outcomes are interpolated, avoiding aggressive extrapolation that is harmful in offline control.
>  - ABC: ABC performs conservative decision-making by using decision supports within a minimal convex hull. How such a decomposition in the convex hull improves conservation is justified theoretically by Proposition 3.8 (Estimation Bound) and Proposition 3.10 (Existence and Uniqueness).
>  - MFRL: In MFRL like CQL and TD3-BC, the conservation is explicitly given as constraints or distribution matching. We would note that conventional MFRL algorithms are not designed for those tasks and suffer from aggressive extrapolation.
>  - MBRL: Similar to MFRL, the conservation should be added to MBRL through external efforts. Because the conventional design of model-based learning does not address such an issue.
>  - BC: In BC, the learning objective is to minimize the prediction difference. There is little we can do to aid conservation.
>
> (3) **Low-Data**: whether a method works in the low-data regime.
>  - ABC: The decision process of ABC only relies on a few examples constituting the minimal convex hull, hence the algorithm performs well under the low-data regime.
>  - MFRL: In MFRL, the black-box value network and policy network can be designed to be sample-efficient.
>  - MBRL: In MBRL, sufficient data is always required to learn an accurate world model.
>  - BC: the performance of BC is highly dependent on the quality of data. It is not designed for the low-data regime.
>
> (4) **Adaptive**: whether the control behavior of a method can be adjusted according to additional constraints as clinical guidelines without modification or re-training.
>  - ABC: by changing reference examples, i.e., the decision corpus, during test time inference, ABC can seamlessly perform different types of decision-making according to user specifications.
>  - MFRL: In MFRL, when new data is used, a new value network and policy network need to be re-trained.
>  - MBRL: In MBRL, the world model construction is independent of the data, hence the decisions can be adaptive by changing a new planning algorithm on top of the world model. No model re-training is needed.
>  - BC: In BC, a new model needs to be trained with a specified type of decision corpus.
>
> (5) **Reward-Free**: availability of extension to the strictly batched imitation setting where rewards are unavailable.
>  - ABC: We have shown the key insight of making decisions through belief space similarity can be extended to the settings without reward signals, as discussed in Appendix C.
>  - MFRL: In MFRL, the Q-values can not be calculated without the reward function.
>  - MBRL: In MBRL, the planning algorithms do not have a clear objective to optimize without reward signals.
>  - BC: BC is not be affected by the absence of reward signals, because it does not need the reward to learn its policy.
>
> ---
> Should there be any additional questions or concerns, we are more than willing to provide further explanations.

---

> > ### Comment · Reviewer_izcS · 2023-08-14
> > **Raise my score**
> >
> > I thank the reviewers for their detailed response to all of my questions. I thoroughly read the other reviewers' feedback and the authors' response. In general, I like the neat idea of finding the minimal hull. However, the authors also admitted that their method primarily focuses on high-stake decision making process, instead of the conventional offline reinforcement learning setting. That is why I hesitated to raise the score to 6 rather than 7 or above.

---

> > > ### Author Response · Authors · 2023-08-15
> > > **Specificity as Asset**
> > >
> > > We sincerely thank you for taking the time to review our manuscript and for your thoughtful consideration of the feedback provided by other reviewers, as well as our responses.
> > >
> > > We acknowledge and respect your perspective regarding the primary focus of our method on high-stake decision-making scenarios. We believe that this specificity can be an asset and unique contribution, as it addresses a crucial area within the broader domain. While previous works on offline RL have primarily focused on efficient learning with conservation, the accountability of offline decisions remains largely unexplored despite its importance. We believe both of those topics are important and warrant their individual study to make scientific progress.
> > >
> > > In critical domains like healthcare, it's vital that decisions are based on supportive evidence. For instance, when a patient is treated in a certain manner, it should be based on the successful outcomes of previous patients with comparable conditions who received the same treatment. The ability to trace the supportive basis of decisions enhances the process of policy reasoning and debugging, thereby improving the trustworthiness of decision-making systems. On the opposite, the supportive evidence is less important in the setting of robotics usually studied by the offline-RL community, which motivates us to select the *interpretability and explainability* rather than reinforcement learning as our primary area.
> > >
> > > We believe the clarity of our manuscript is enhanced with revisions addressing your insightful comments. And our novel approach, coupled with the expansive applications of accountability in batched control tasks, holds the potential to make a meaningful contribution to the community.
> > >
> > > Once again, we deeply appreciate the attention and thoroughness you have provided throughout the review process.

---

### Official Review · Reviewer_9Kjz · 2023-07-07

**Soundness:** 4 excellent
**Presentation:** 3 good
**Contribution:** 3 good
**Rating:** 7
**Confidence:** 3

**Summary:**

The paper presents the Accountable Batched Controller (ABC) based on the example-based explanation framework as a solution for offline control in responsibility-sensitive applications. Through experiments on simulated and real-world tasks, the method shows accountability, conservation, and adaptability.

**Strengths:**

The paper proposed a novel method for accountable batched control with decision corpus, theoretically proved the existence and uniqueness of the decomposition under mild conditions, and conducted solid experiments to verify the effectiveness and desired properties of the method.

The paper is clear-presented and well-organized.

**Weaknesses:**

To further improve, more experiments on real-world control tasks are needed.

**Questions:**

More in-depth analysis of Table 2 may help gain deeper insight regarding the suitability for low-data regimes.

Why is section 5.1 highlighting P1-P3?

**Limitations:**

As the authors also mentioned, the current accountable batched control method is limited to low-dimensional control tasks and may not perform as well in high-dimensional control systems, limiting its potential contribution.

---

> ### Author Rebuttal · Authors · 2023-08-08
>
> We thank the reviewer for the time and effort in reviewing our paper. We will respond to each point in turn:
>
> ---
> ### 1. Additional Analysis on Table 2
>
> We would start by providing **more empirical studies** extending the previous Table 2:
>
> _The cumulative reward of each method is reported. Additional experiments are repeated with 5 seeds. **Higher is better.**_
>
> | Task              | Low-Data            | Mid-Data            | Rich-Data           |
> |-------------------|---------------------|---------------------|---------------------|
> | 1NN               | -557.07 ± 256.64    | -690.49 ± 152.59    | -512.71 ± 131.2     |
> | kNN               | -849.45 ± 91.23     | -670.51 ± 321.09    | -645.72 ± 220.33    |
> | kNN + Belief      | -659.17 ± 219.76    | -525.58 ± 436.56    | -534.02 ± 568.47    |
> | ABC w/o Belief    | -302.55 ± 426.39    | -173.84 ± 245.85    | -130.24 ± 184.12    |
> | ABC               | **-1.39 ± 1.39**    | **-1.25 ± 0.40**    | **-0.6 ± 0.08**     |
> | BC                | -422.77 ± 409.51    | -225.32 ± 340.83    | -126.74 ± 280.73    |
> | TD3               | -4.1 ± 2.76         | -11.95 ± 4.68       | -15.27 ± 6.46       |
> | CQL               | -793.89 ± 206.0     | -889.85 ± 291.99    | -805.49 ± 578.75    |
> | TD3-BC            | -844.95 ± 170.93    | -781.8 ± 337.11     | -821.02 ± 587.77    |
> | TD3-BC-Recurrent  | -82.92 ± 115.79     | -41.92 ± 59.28      | **-0.45 ± 0.57**    |
> | MPC               | **-1.5 ± 0.43**     | **-1.34 ± 0.15**    | -1.41 ± 0.26        |
> | Data-Avg-Return   | -307.81 ± 387.53    | -245.54 ± 338.65    | -208.81 ± 272.84    |
>
> In this updated Table, we have
>
> (1) included ablation studies (kNN+Belief, ABC w/o Belief)
>
> (2) included additional baselines in offline-RL
>
> Several conclusions can be drawn from the updated table:
>
> - (1) [High-Performance] Compare to all of the methods including black-box algorithms and the accountable baselines, we find** ABC is able to achieve high performance in all settings**.
>
> - (2) [Efficacy under Low-Data Regime] Compare ABC with the baselines, we observe the superiority of ABC especially under the low-data regime. In such a setting, **ABC is able to effectively solve the problem while many other methods suffer from higher instability**.
>
> - (3) [Ablation Studies] In addition to the comparison between ABC and kNN, we **additionally experiment** with kNN that works in the belief space, we find it improves the performance of kNN, demonstrating the **effectiveness of leveraging the belief space** in accountable decision making. Moreover, we demonstrate the **effectiveness of the minimal convex hull decomposition**, which is another algorithmic design, through the experiment of ABC w/o Belief (i.e., kNN with minimal convex hull decomposition). We find the results are better than the original kNN, yet significantly worse than ABC.
>
> - (4) [Instability of Offline-RL Algorithms] Conventional offline-RL algorithms focus only on the conservatism during learning from offline decision corpus in MDP tasks, hence **they fail for efficient learning in our partially observable tasks with high stochasticity**. To make those algorithms stronger, we additionally implemented a recurrent module for the TD3-BC algorithm, and compare it to ABC. We find those offline-RL algorithms **suffer from instability issues during learning and are hard to converge** to a well-performing policy. We provide an additional analysis of their learning process in Appendix D.7.
>
> - (5) [Property of Conservation] We would like to note the property of conservation is demonstrated through the _offline nature_ of those benchmark tasks.
> To enhance the clarity of our representation, we would refer to the **added ablation study that could further help demonstrate the importance of conservation**: through the comparison between kNN and ABC w/o Belief (as they both work in the original input space) or kNN+Belief and ABC (as they both work in the belief space), the performance gain of the ABC w/o Belief and ABC over counterparts are actually based on the conservative property introduced by the minimal convex hull.
>
> ### 2. More tasks
>
> To address the reviewer's concern on more challenging tasks, we refer to additional experiments in Appendix F.3, F.4, F.5, where we demonstrate the scalability of ABC and the potential of combining ABC with black-box policies. From such a perspective, ABC can be used as a post-hoc interpretation mechanism for any given black-box algorithms
>
> To be specific, we additionally experimented on the LunarLander-Continuous environment and the BipedalWalker environment. In those experiments, we find that the dimensionality of states is not a critical issue, but the increase in the action dimensions can be more challenging — it originates from the uniform sampling over the action space in our Algorithm. To address such a difficulty, we investigate the potential of integrating ABC with black-box samplers in F.4.
>
> We can observe from the results that ABC can work both in isolation or combined with black-box policies. ABC can be used as a plug-in to add accountability to black-box controllers in a post-hoc manner. In high-dimensional control tasks, uniform sampling can be inefficient and black-box samplers can alleviate such a difficulty. This could potentially be a promising direction for future research.
>
> ---
> Should there be any additional questions or concerns, we are more than willing to provide further explanations.

---

> > ### Comment · Reviewer_9Kjz · 2023-08-18
> > **Thank you for the response**
> >
> > I want to thank the authors for the detailed response, I have considered them, and I maintain my original score.

---

> ### Author Response · Authors · 2023-08-15
> **Further Discussions and Feedback Welcome**
>
> We deeply appreciate the insights you've shared during the review process. Following our revisions and previous responses, we are genuinely curious if we have adequately addressed the concerns you raised.
>
> Should there be any leftover questions, concerns, or areas you feel need more clarification, please do not hesitate to let us know. We greatly respect your insights and stand ready to make any additional refinements based on your feedback.

---

### Official Review · Reviewer_fhj3 · 2023-07-09

**Soundness:** 2 fair
**Presentation:** 2 fair
**Contribution:** 3 good
**Rating:** 5
**Confidence:** 4

**Summary:**

I found the paper somewhat hard to read and understand, so here I’ll present a summary that’s quite different from the author’s presentation.

In offline RL, or other settings where there is a performance metric to optimize, we can consider two simple baselines:
1. Nearest neighbors: For each action, find the most similar transition(s) in the dataset, and use those to estimate the value of the action, then take the action estimated to be best.
2. Supervised learning: Train a model to predict the value of each action, and use that to estimate the value of actions.

The advantage of (1) is that we get a notion of explainability (visualize the nearest neighbors that were used to estimate action value), but the disadvantage is that it does not work well (because similarity in the input space may not mean that decision-making will be similar). The advantage of (2) is that it works better, but is less explainable. So the first idea is that we can get the best of both worlds by still using (2) to train a model, but then use the embeddings (i.e. the activations before the final linear layer) as inputs for a nearest neighbors approach.

However, this can still have problems: in particular, for a new test point, the nearest neighbors may all be very tightly clustered but far away from the test point. Ideally, in such a situation, we would find nearby points in a variety of _different_ directions, and average them, so that our estimates are interpolations rather than extrapolations in the embedding space. So, instead of finding the nearest neighbors in our dataset, we find a minimal set of points from the dataset such that the current embedding falls within the convex hull of those points (or, if no such set exists, the embedding is as close as possible to the convex hull). We automatically discard any actions that are far away from the best convex hull, since they are likely OOD. This gives the author’s method: ABC.

(In the paper’s presentation, the embeddings are called “beliefs”.)

The authors test their method in a variety of settings:

1. Heterogeneous Pendulum: Similar to classic Pendulum, except that there is a 50% chance for the action effects to be swapped.
2. Maze: A 2D setting with a wall separating the start and goal states, with two openings in the wall.
3. Ward: Healthcare task, in which the task is to predict whether or not to use an oxygen therapy device.

In heterogeneous Pendulum, the authors show that ABC performs slightly better than model-free RL and model predictive control, and much better than other baselines. They also show the effect of ablating $\epsilon$, the hyperparameter that controls which actions are considered OOD.

In Maze, the authors collect data from a variety of different behavioral policies, which have to be composed together to solve the task, and show that ABC is capable of this. There are two different ways to solve the task, corresponding to the two openings in the wall. The authors show that ABC can show both methods of solving the task, and that when visualizing the points forming the minimal convex hull for the resulting actions, they can be attributed to the behavior policies that used the same hole in the wall to solve the task. They also show that by increasing the proportion of different behavioral policies, you can control which of the two solutions ABC is more likely to use.

On Ward, the authors show that ABC performs on par with behavior cloning (BC) using a multilayer perceptron, and performs better than k-nearest-neighbors and BC using a linear model.

**Strengths:**

1. Once I understood the idea, I found it simple and intuitive, with a clear story about why it should be helpful.
2. The application of machine learning to healthcare is important, and accountability and conservatism are important properties to ensure in such a setting.
3. There are a variety of experiments demonstrating the claimed properties of the method.

**Weaknesses:**

**Properties of ABC**

The authors list five properties that ABC satisfies. I agree with the author’s points that ABC works better with low data (at least relative to kNN) and that ABC can be used in the reward-free setting (at least for continuous actions spaces). However, I’m not convinced of the other three properties:

1. Conservatism: The authors claim that ABC is conservative, I believe because they filter out actions that have a belief corpus residual that is too high. While I think the authors are probably correct, I don’t think their experiments show it: in all of the experiments that compare against baselines, black-box methods perform about the same as ABC, even though black-box methods are not normally “conservative”.
2. Accountable: The authors claim that ABC is accountable because for any action taken by ABC, we can identify data points in the training dataset that make up the convex hull that determined that particular action, and show those to the user. However, there isn’t even a qualitative evaluation of how useful such explanations are. The closest is Figure 5, which visualizes the belief corpus as points on a 2D grid whose axes are uninterpretable (belief dimensions 1 and 2) relative to the test data point, but looking at that figure I do not feel like I have understood very much about ABC’s decision in that setting.
3. Adaptive: To show that the ABC is adaptive, the authors perform an experiment in which they change the composition of the dataset on which ABC is trained, and show that this affects ABC’s behavior. But by this standard, essentially all algorithms are adaptive, including the baselines they compare against (e.g. behavior cloning, which they say is not adaptive in Table 1). It’s not clear why this is a unique advantage of ABC.

(Incidentally, on accountability, the author’s technique is extremely similar to the technique of presenting maximally activating dataset examples to explain neuron activations, a common technique for explainability in supervised learning.)

**Additional comparisons**

I would like to see the authors compare ABC to the first method in my summary, i.e. training a model to predict value / actions, and then using k-nearest-neighbors on the embeddings (activations before the final linear layer). This can be thought of either as a baseline, or as an ablation (as an ablation, it corresponds to ABC without the convex hull aspects). This would be helpful in understanding the effects of the various design decisions the authors make.

If performing an experiment on accountability, then I’d like to see a comparison to the dataset examples technique applied to the kNN-on-embeddings model discussed above.

**Disagreement with Section 5.2 claim**

Section 5.2 notes that there are two hyperparameters: “the number of uniformly sampled actions and the threshold”. It claims that these can be unified into a single hyperparameter, the _effective action size_. However, the experiment doesn’t support this: it simply sets the number of sampled actions (which we’ll call $n_A$) to 100, and then shows the effect of varying the percentile threshold $\epsilon$. The experiment that should be run would be to use a variety of settings of _both_ hyperparameters, and then check whether runs with similar effective action sizes $\epsilon \times n_A$ have similar performance: if so, then it would be justified to only think about the effective action size. However, my guess is that this will not be the case.

**Minor issue with the theory**

(Note: set notation doesn't seem to be working below)

Proposition 3.10 is false because of the requirement that the convex hull contain $d_b + 1$ examples. For example, suppose $d_b = 2$, $b_t = [1, 0]$, and $\mathcal{D} = \{ [7, 0], [3, 0], [-1, 0] \}$. Note that $b_t = 0.5 * [-1, 0] + 0.5 * [3, 0]$, and so if we have $\mathcal{C} = \mathcal{D}$, then $b_t \in \mathcal{CB(C)}$, and so $r_{\mathcal{C}}(b_t) = 0$ as required by Proposition 3.10. Definition 3.9 requires the minimal corpus subset $\tilde{\mathcal{C}}(b_t)$ to have 3 elements, and which means that $\tilde{\mathcal{C}}(b_t) = \mathcal{D}$. However, the decomposition on the minimal hull is not unique, since we have both $b_t = 0.5 * [-1, 0] + 0.5 * [3, 0]$ as well as $b_t = 0.75 * [-1, 0] + 0.25 * [7, 0]$, contradicting Proposition 3.10.

The issue is that you require the convex hull to contain $d_b + 1$ examples. If you remove that restriction, then in the example above $\tilde{\mathcal{C}}(b_t) = \{ [-1, 0], [3, 0] \}$ and then the decomposition is unique, as desired.

(I believe your current proof would also work if you remove the restriction. Currently, it doesn’t work because you remove an element from $\tilde{\mathcal{C}}(b_t)$ and call that a contradiction, but it is actually not a contradiction because the newly created set no longer has $d_b + 1$ elements.)

**Questions:**

**Overall view and suggestions for the authors**

I quite like the idea in this paper, but currently I think the evaluation and presentation of the idea are not good enough, and so I am recommending rejection. However, I think there is the seed of a good paper here, and would likely be quite excited about a version of the paper that looked more like:

1. Discussing embeddings as a useful way to get a nice structured representation, perhaps considering kNN-on-embeddings as the baseline.
2. Identifying linear interpolation in the minimal convex hull as an alternative to the kNN decision criterion.
3. Conducting a series of experiments that demonstrate the value of the convex hull idea, focusing particularly on questions like: (a) Are belief corpus residuals better at OOD detection than nearest-neighbor distances? (b) Does linear interpolation in a convex hull lead to better performance than kNN on embeddings? (c) Do the points in the convex hull provide a better explanation of the selected action than the k nearest neighbors in embedding space? I think it is quite plausible that convex hulls do better on all of these metrics, but the current experiments don’t show it.

**Note on confidence**

I’ve selected a confidence of (4) below, but I want to note that I am not very familiar with related literature, and in particular I know very little about accountability in the healthcare setting. As a result, I cannot evaluate (1) the originality of the work (maybe convex hulls have been explored before), and (2) whether the authors compared to state of the art techniques.

**Questions**

I’m interested in responses to any of the weaknesses I listed above, but in addition, I have some questions on specific details:

1. Why do you require that the minimal corpus subset $\tilde{C}(b_t)$ have $d_b + 1$ elements?
2. In Section 3.5, $\epsilon$ appears to be an absolute threshold for the belief corpus residual, but in Section 5.2, it appears to be a percentile for the belief corpus residual. Which is it?
3. In the Maze environment, what is the performance measure for the behavioral policies that make up your dataset? Is it cumulative reward on the final task (i.e. going from (0,0) to (16,0)) or cumulative reward on each individual task (i.e. going from (0,0) to (8, 16), going from (0, 0) to (8, 8), etc), or something else entirely?

**Limitations:**

The weaknesses listed above are not present in the paper. Suggestions for improvements are in the previous sections.

The paper applies ABC to discrete settings, and also says that it can work in reward-free settings, but the idea for reward-free settings only works for continuous action spaces, not discrete ones. This is mostly not a big deal since the reward-free setting is not currently a major focus of the paper.

---

> ### Author Rebuttal · Authors · 2023-08-08
>
> We thank the reviewer for the time and effort in reviewing our paper. We will respond to each point in turn:
>
> ---
>
> ### 1. Properties
>
> - **Conservatism**. We demonstrate the property of Conservation through our experiments provided in Appendix F2. We explicitly visualize the behaviors of ABC under different degrees of conservation.
>
> - **Accountability**. ABC is accountable because it provides reference examples in making decisions. The high-level insight follows example-based explanations in XAI.  In our work, we put a special focus on sequential decision-making problems rather than prediction tasks in the XAI literature.
>
> - **Adaptivity**. We recognized our previous description of adaptivity could be misleading. In fact, we wish to demonstrate the property of on-the-fly adaptivity of ABC that is correlated to its property of accountability: as it links decisions to reference training data, filtering out those unwanted training data during deployment is a much easier way than training a new model like BC and other algorithms to perform a special type of decision under user specification.
>
> ### 2. Ablation studies, and additional baselines
>
> To address the reviewer's concern about empirical evaluation, we implemented two more variants of ABC and kNN as baselines, including
> - **kNN+Belief**: it uses the learned latent space for decision-making when applying kNN. This study could be regarded as the ablation study of _ABC w/o Minimal Hull_.
> - **ABC w/o Belief**: it uses the original input space for decision-making, but applies the minimal hull decomposition. This study could be regarded as the ablation study of _kNN + Minimal Hull_.
>
> Results are provided in the following extended Table 2:
>
> _The cumulative reward of each method is reported. Additional experiments are repeated with 5 seeds. **Higher is better.**_
>
> | Task              | Low-Data            | Mid-Data            | Rich-Data           |
> |-------------------|---------------------|---------------------|---------------------|
> | 1NN               | -557.07 ± 256.64    | -690.49 ± 152.59    | -512.71 ± 131.2     |
> | kNN               | -849.45 ± 91.23     | -670.51 ± 321.09    | -645.72 ± 220.33    |
> | kNN + Belief      | -659.17 ± 219.76    | -525.58 ± 436.56    | -534.02 ± 568.47    |
> | ABC w/o Belief    | -302.55 ± 426.39    | -173.84 ± 245.85    | -130.24 ± 184.12    |
> | ABC               | **-1.39 ± 1.39**    | **-1.25 ± 0.40**    | **-0.6 ± 0.08**     |
> | BC                | -422.77 ± 409.51    | -225.32 ± 340.83    | -126.74 ± 280.73    |
> | TD3               | -4.1 ± 2.76         | -11.95 ± 4.68       | -15.27 ± 6.46       |
> | CQL               | -793.89 ± 206.0     | -889.85 ± 291.99    | -805.49 ± 578.75    |
> | TD3-BC            | -844.95 ± 170.93    | -781.8 ± 337.11     | -821.02 ± 587.77    |
> | TD3-BC-Recurrent  | -82.92 ± 115.79     | -41.92 ± 59.28      | **-0.45 ± 0.57**    |
> | MPC               | **-1.5 ± 0.43**     | **-1.34 ± 0.15**    | -1.41 ± 0.26        |
> | Data-Avg-Return   | -307.81 ± 387.53    | -245.54 ± 338.65    | -208.81 ± 272.84    |
>
> We additionally compare against CQL and TD3-BC in Section 5.1. Not surprisingly, we find the performance of the original CQL and TD3-BC on Pendulum-Het are poor. This is because they are not designed for partial observable tasks with high stochasticity.
> Therefore, we further implemented an improved version of the TD3-BC with a recurrent context encoding module [cf. Meta-Q-Learning]. In those experiments, we find the learning process of Offline-RL is not stable, leading to a large variance in the policy quality — we have observed a similar problem in our previous MFRL baseline, and reported it with details in Appendix D.7.
>
>
> ### 3. Hyper-parameters
>
> We would agree with the reviewer that using a smaller sampling size with a larger quantile number is less preferred in comparison with using a larger sampling size with a smaller quantile number, this is because the latter will lead to a more accurate estimation and more conservative behavior.
>
> We have updated our manuscript accordingly.
>
> ### 4. Theory
>
> In the counter-example raised by the reviewer, the dimension of the belief space diminished to $1$-dim, instead of 2-dim, for the mentioned specific convex hull decomposition.
> The key issue is indeed **the dimension of the belief space should be more clearly defined**. In this example, the minimal hull should not contain three points, because its hyper-volume (i.e., length, in this 2-D example) is not minimized.
> We have updated the dimension of the search space from **$d_b+1$** to **maximally $d_b +1$** to enhance clarity.
>
> ### 5. Threshold
>
> To make our methodology part clear, we use $\epsilon$ as a constant in Sec. 3.5. In practice, such a constant can be implemented through quantile thresholding (Sec.5.2). We use the same notation to emphasize that this quantile number controls the threshold. We updated the notation in Sec. 5.2 to $\epsilon(q=0.3), \epsilon(q=0.5)$, etc. to enhance the clarity.
>
> ### 6. Terminology of _Reward-Free_
>
> In our context, the reward-free indicates the **strictly batched imitation** settings where reward information is not accessible. Different from the normal batched control setting where an offline dataset containing $(o_t, a_t, r_t)$ is available, the reward-free setting can only leverage a dataset that is composed of $(o_t, a_t)$.
>
> ---
> Should there be any additional questions or concerns, we are more than willing to provide further explanations.

---

> > ### Comment · Reviewer_fhj3 · 2023-08-14
> > **Raising my score**
> >
> > Thanks for the response! It has addressed most of my concerns, and I am raising my score from 3 to 5 (and contribution from 2 to 3).
> >
> > My main remaining concerns are:
> >
> > 1. While it is true that with the authors’ method it is possible to identify the data points in training that affect the action taken, it is not clear to me how much accountability this provides.
> > 2. I find it hard to square the results of Appendix F2 and the failure of the kNN + embeddings method with the success of black box methods, which makes me think I’m misunderstanding something about the paper.
> > 3. As mentioned in the review, I find the presentation of the paper quite confusing.

---

> > > ### Author Response · Authors · 2023-08-14
> > > **Follow-up Response**
> > >
> > > Thank you for the continued evaluation and the follow-up questions. We hope our explanation below could be helpful in addressing your remaining concerns:
> > >
> > > ### 1. Accountability through reference examples.
> > >
> > > ABC offers a novel example-based approach to interpretable policy learning. As has been shown in [1], **human subjects in fact find example-based explanations more insightful than explanations based on feature importance**, especially for human-machine cooperative tasks.
> > >
> > > To see how example-based interpretability (i.e., accountability) can be more helpful and distinguish from feature-based interpretability in decision-making, we will illustrate with an example of cancer treatment involving high-risk options like Radiotherapy and Chemotherapy.
> > >
> > > **Conventional Interpretable-RL is for Model Understanding**
> > >
> > > Existing interpretability methods in RL, such as feature saliency[2], input importance[3], and converting black-box models into interpretable formats[4], primarily helps users understand how models arrive at decisions. Using these methods, users can understand **how decisions correlate with specific features**. In the context of cancer treatment, such interpretations might attribute decisions to certain biomarker levels. This interpretability facilitates debugging and refining model decisions. For instance, if the policy put focus on some unnecessary or causally unrelated features, doctors and experts can improve the policy learning by removing those inputs[3,5].
> > >
> > > We would like to note the focus in such a case is to **debug and improve the model’s decision**.
> > >
> > >
> > > **Accountability Benefits Human-AI Cooperation**
> > >
> > > However, **why** a policy generates the decisions is remain unclear (e.g., what is the decision support?). Even with the above type of interpretability, people may still wonder why a certain biomarker should be able to determine the treatment plan. And knowing there are successful cases when similar patients receive the same treatment plan will be beneficial for those non-experts (e.g., patients) to understand the process. (and importantly, be optimistic about the outcome.)
> > >
> > > ABC shifts the focus to understanding why policies decide as they do, especially when the reason is non-obvious. In critical applications, like the aforementioned cancer treatment, understanding the why is crucial. ABC enhances human-AI cooperation by offering reference examples, which are **more intuitive for humans**, aiding them in complex decision-making. ABC achieves this by mapping examples to the belief space that is linearly dependent on the outcome and identifying supporting examples through a minimal convex hull decomposition in such a space.
> > >
> > >
> > >
> > > ### 2. Integrating ABC with Black-Box Models
> > >
> > > In Appendix F.2, we highlighted that ABC can be used as a post-hoc interpretation module by combining it with any black-box policies in decision-making. This is because of the fact that for any given transition history and action, ABC is able to find the corresponding minimal convex hull decomposition, and therefore find the most representative reference examples for executing such a given action. In high-dimensional tasks where uniform sampling from the action space can be inefficient, leveraging a black-box model as the sampler can achieve a good balance between accountability and performance.
> > >
> > >
> > > ---
> > > **References**
> > >
> > > [1] Nguyen, Giang, Daeyoung Kim, and Anh Nguyen. "The effectiveness of feature attribution methods and its correlation with automatic evaluation scores." Advances in Neural Information Processing Systems 34 (2021): 26422-26436.
> > >
> > > [2] Mott, Alexander, et al. "Towards interpretable reinforcement learning using attention augmented agents." Advances in neural information processing systems 32 (2019).
> > >
> > > [3] Yujin Tang, Duong Nguyen, and David Ha. Neuroevolution of self-interpretable agents. In Proceedings of the 2020 Genetic and Evolutionary Computation Conference, pages 414–424, 2020.
> > >
> > > [4] Daniel Hein, Steffen Udluft, and Thomas A Runkler. Interpretable policies for reinforcement learning by genetic programming. Engineering Applications of Artificial Intelligence 158–169, 2018
> > >
> > > [5] De Haan, Pim, Dinesh Jayaraman, and Sergey Levine. "Causal confusion in imitation learning." Advances in Neural Information Processing Systems 32 (2019).
> > >
> > >
> > > ---
> > > Thank you again for your consideration and supportive feedback. Should there be any leftover concerns, please let us know and we will do our utmost to address them.

---

> > > ### Author Response · Authors · 2023-08-18
> > > **Follow-up Response on Presentation**
> > >
> > > Dear Reviewer fhj3,
> > >
> > > In response to your feedback regarding the presentation, and to incorporate your valuable comments, we have made a series of updates to our manuscript. We have detailed these changes in the official comment titled "Follow-Up Author Response on Presentation to Reviewer fhj3 and fQBY".
> > >
> > > We hope the reorganized introduction and the method sketch paragraph inspired by your comments could address your concerns about our presentation. We would appreciate it if you could kindly let us know if there were any further questions. In the limited time remaining, we are still eager to do our utmost to address them!
> > >
> > > Regards,
> > >
> > > Authors

---

### Author Rebuttal · Authors · 2023-08-09

We extend our sincere gratitude to all reviewers for their insightful comments, valuable suggestions, time, and efforts in evaluating and improving our paper.

We thank all reviewers for their affirmation of our work’s **novelty** (reviewers: 9Kjz, izcS), **presentation** (reviewers: fhj3, 9Kjz, izcS, fQBY), **evaluation** (reviewers: fhj3, 9Kjz, izcS, qevo), and **importance** (reviewers: fhj3, fQBY, qevo).

----

To address the concerns raised by reviewers, we would respond to each of their questions respectively. Below, as a general response, we aim to outline the **key revisions and additional experimentation conducted by far**:

#### **Supplementary Experimental Evaluation**
1. **(Table 1 in attached PDF)** We conducted additional ablation studies of
  - kNN + Belief  (i.e., ABC w/o Minimal Convex Hull Decomposition)
  - ABC w/o Belief (i.e., kNN + Minimal Convex Hull Decomposition)
2. **(Table 2 in attached PDF)** We provide additional offline-RL baselines, including
  - CQL [1]
  - TD3-BC [2]
  - TD3-BC-Recurrent that improves TD3-BC using a recurrent module in POMDP tasks [3,4].
3. **(Figure 1 in attached PDF)** We highlight the controllable conservative behavior of ABC by varying the quantile number.
4. **(Table 3 in attached PDF)** We demonstrate the scalability of ABC by providing results on the LunarLanderContinuous environment.
5. **(Table 4 in attached PDF)** We experiment with the BipedalWalker environment to showcase how to integrate ABC with black-box controllers to add accountability.

#### **Revised Manuscript for Clarity**
1. We have revised the terminology used in Proposition 3.10 to eliminate any ambiguity regarding the dimension restriction of the minimal convex hull.
2. We have emphasized that the primary focus of ABC extends beyond the problem of offline RL. While offline RL possesses only the conservation property, ABC has five distinct properties that are all crucial for accountable batched control tasks.
3. In Sec.3.5 and Sec.5.2, we've employed distinct symbols to represent the constant, using $ \epsilon$, and the constant as a function of a given quantile threshold, denoted as $ \epsilon(q) $, to enhance clarity.
4. We explained the 5 desired properties more explicitly in the related work section, illustrating why compared baselines may not satisfy each of those properties.


----

We hope our clarification and additional empirical studies could address the concerns raised by reviewers. Should there be any leftover questions, please let us know and we will make every effort to address them during the subsequent discussion period.


---
**_Refrences_**

[1] Kumar, Aviral, et al. "Conservative q-learning for offline reinforcement learning."

[2] Fujimoto, Scott, and Shixiang Shane Gu. "A minimalist approach to offline reinforcement learning."

[3] Ni, Tianwei, Benjamin Eysenbach, and Ruslan Salakhutdinov. "Recurrent model-free rl can be a strong baseline for many pomdps."

[4] Fakoor, Rasool, et al. "Meta-q-learning."

---

### Author Response · Authors · 2023-08-18
**Follow-Up Author Response on Presentation to Reviewer fhj3 and fQBY (Part 1/3)**

### General Response on Presentation
In this work, we address the important yet previously underexplored problem of accountability in decision-making. To bridge the gap between example-based explanation in XAI and our problem setting, it is necessary to introduce multiple **novel yet non-trivial formal definitions**:
- Definition 3.1 Corpus Subset,
- Property 3.2 Belief Space Linearity,
- Definition 3.5 Belief Corpus Subset,
- Definition 3.6 Belief Corpus Convex Hull,
- Definition 3.7 Belief Corpus Residual,
- Definition 3.9 Minimal Hull and Minimal Corpus Subset

Those formal definitions might seem obscure at first glance, yet they are **crucial and necessary**. This is because only with those formal definitions, we are able to provide the two key propositions
-  Proposition 3.8: Estimation error bound of value estimation
-  Proposition 3.10: Existence and uniqueness of the decomposition

the former proposition provides a theoretical guarantee of the value estimation, while the latter proposition aids the stability and robustness of the proposed method.

**We recognized during the rebuttal period that providing a high-level roadmap, as of the proof sketch in technical literature, to anchor the flow of our method, can be helpful.** Therefore, we integrated the advice from our reviewers and made the following revision:
1. Re-write the introduction, with more concrete examples and clearer motivations
2. Add a roadmap section before formally introducing the method
3. Re-organize the related work section, highlighting the connection and difference from Offline-RL

---
### **1. Re-organized Introduction** (new contents highlighted with shaded boxes)
In recent years, batched control that uses pre-collected data to generate control policies has gained attention due to its potential to reduce the costs and risks associated with applying control algorithms in real-world systems, which is especially advantageous in situations where real-time feedback is challenging or expensive to obtain.
> However, in many critical real-world applications such as healthcare, the challenge is beyond enhancing policy performance. It requires the decisions made by learned policies to be transparent, traceable, and justifiable. Yet those essential properties, summarized as _Accountability_, are left largely unaddressed by existing literature.

> In our context, we use _Accountability_ to indicate the existence of a _supportive basis for decision-making_. For instance, in tumor treatment using high-risk options like radiotherapy and chemotherapy, the treatment decisions should be based on the successful outcomes experienced by previous patients who share similar conditions that were given the same medication.
> Another concrete illustrative example is the allocation decisions of ventilator machines. The decision to allocate a ventilator should be accountable, in the way that it juxtaposes the potential consequences of both utilization and non-utilization and provide a reasonable decision on those bases.
> In those examples, the ability to refer to existing cases that support current decisions can enhance reliability and facilitate reasoning or debugging of the policy.

> To advance offline control towards real-world responsibility-sensitive applications, five properties are desirable:
> - (P1) Controllable **Conservation**: this ensures the policy learning performance by avoiding aggressive extrapolation.
> - (P2) **Accountability**: as underscored by our prior examples, there is a need for a clear basis upon which decisions are made.
> - (P3) Suitability for **Low-Data** Regimes: given the frequent scarcity of high-stake decision data, it’s essential to have methods that perform well with limited data.
> - (P4) **Adaptivity** to User Specification: this ensures the policy can adjust to changes, like evolving clinical guidelines, allowing for tailored solutions
> - (P5) **Flexibility** in Strictly Batched Imitation: this property ensures a broader applicability across various scenarios and data availability.

> To embody all these properties, we need to venture beyond the current scope of literature focused on conservative offline learning. In our work:
> - Methodologically, we introduce the formal definitions and necessary concepts in accountable decision-making. We propose the Accountable Batched Controller (ABC), which makes decisions according to a decomposition on the basis of the representative existing decision examples.
> - Theoretically, we prove the existence and uniqueness of the decomposition under mild conditions, guiding the design of our algorithms.
> - Practically, we introduce an efficient algorithm that takes all the aforementioned desired properties into consideration and circumvented the computational difficulty.
> - Empirically, we verify and highlight the desired properties of ABC on a variety of batched control tasks, including five simulated continuous control tasks and one real-world healthcare dataset.

---

> ### Author Response · Authors · 2023-08-18
> **Follow-Up Author Response on Presentation (Part 2/3)**
>
> ### **2. Method Sketch: A Roadmap** (this paragraph is added to the beginning of Section 3)
>
>
> _We would like to extend our appreciation to reviewer fhj3 for inspiring the following paragraph._
>
> > The high-level core idea of our work is to introduce an example-based accountable framework for offline decision-making, such that the decision basis can be clear and transparent.
>
> > To achieve this, a naive approach would be to leverage the insights of Nearest Neighbors: for each action, this involves finding the most similar transitions in the offline dataset, and estimating the corresponding outcomes. Nonetheless, a pivotal hurdle arises in defining **similarity**, particularly when taking into account the intricate nature of trajectories, given both the observation space heterogeneity and the inherent temporal structure in decision-making. Compounding such a challenge, another difficulty arises in identifying the most **representative** examples, and integrating the pivotal principle of **conservation**, which is widely acknowledged to be essential for the success of offline policy learning.
>
>
> > Our proposed method seeks to address those challenges.
> > We start by introducing the basic definitions to support a formal discussion of accountability: in Definition 3.1, we introduce the concept of Decision Corpus.
>
> > To address the **similarity** challenge, we showcase a nice linear property (Property 3.2) generally exists (Remarks 3.3 & 3.4) when working in the belief space (Definition 3.5). This subsequently leads to a theoretical bound for estimation error (Proposition 3.8);
>
> > To address the **representative** challenge while obeying the principle of **conservation**, we underscore those examples that span the convex hull (Definition 3.6) and introduce the related optimization objective (Definition 3.7 & 3.9). In a nutshell, the intuition is to use a minimal set of **representative** training examples to **encapsulate** test-time decisions. Under mild conditions, we show the solution would exist and be unique (Proposition 3.10).
>
> > Finally, we outline the optimization procedures and provide pseudo-code at the section’s close.
>
>
> ---
> ### **3. Extended Related Work (in Appendix)** (added to the discussion section on Offline-RL)
>
> > In offline-RL, both model-based and model-free approaches leverage black-box approximators. As a consequence, the pursuance of accountability can not be achieved through those conventional algorithms.
>
> > We would like to note that, although ABC also studies the control problems under the offline setting, its focus goes beyond the conservative efficient learning objective in offline-RL literature. As we have demonstrated in the **Table** below, ABC has five distinct properties that are all crucial for accountable batched control tasks:
>
> > - **P1 (Conservation)**
> > - **P2 (Accountability)**
> > - **P3 (Low-Data Requirement)**
> > - **P4 (Adaptivity)**
> > - **P5 (Reward-Free)**
>
> _**Table**: ABC is distinct as it satisfies 5 desired properties._
> | Method / Property | Conservation | Accountable | Low-Data | Adaptive | Reward-Free |
> |---|---|---|---|---|---|
> | Model-Free RL | ✅ | ❌ | ✅ | ❌ | ❌ |
> | Nearest Neighbor | ❌ | ✅ | ❌ | ✅ | ✅ |
> | Model-Based RL | ❌ | ❌ | ❌ | ✅ | ❌ |
> | Behavior Clone | ❌ | ❌ | ❌ | ❌ | ✅ |
> | ABC | ✅ | ✅ | ✅ | ✅ | ✅ |
>
>
> > Below, we further explain those properties and corresponding methods in turn. For each of the properties, we start with introducing the definitions, followed by comparisons among ABC and MFRL, MBRL, and BC. The discussion on MFRL and MBRL include the offline-RL algorithms.
>
> > (1) Accountability: the decision-making process is traceable, and the decisions can be supported by concrete examples in the offline dataset.
> > - ABC: The decision-making process of ABC is supported by a corpus subset from the offline dataset, hence all the decisions are transparent and traceable.
> > - MFRL: In MFRL, a black-box value network and black-box policy network are learned with the offline dataset. There is no decision support for the black-box policies.
> > - MBRL: In MBRL, a black-box world model optimized with the offline dataset is used as a proxy of the actual dynamics, and planning algorithms are then applied to such a black-box model to make decisions. Those decisions are not supported by explicit references.
> > - BC: In BC, a black-box policy is learned through supervised learning. The output of such a policy is hard to be linked with specific training examples.

---

> ### Author Response · Authors · 2023-08-18
> **Follow-Up Author Response on Presentation (Part 3/3)**
>
> > (2) Conservation: estimations of decision outcomes are interpolated, avoiding aggressive extrapolation that is harmful in offline control.
> > - ABC: ABC performs conservative decision-making by using decision supports within a minimal convex hull. How such a decomposition in the convex hull improves conservation is justified theoretically by Proposition 3.8 (Estimation Bound) and Proposition 3.10 (Existence and Uniqueness).
> > - MFRL: In MFRL like CQL and TD3-BC, the conservation is explicitly given as constraints or distribution matching. We would note that conventional MFRL algorithms are not designed for those tasks and suffer from aggressive extrapolation.
> > - MBRL: Similar to MFRL, external efforts should add conservation to MBRL. Because the conventional design of model-based learning does not address such an issue.
> > - BC: In BC, the learning objective is to minimize the prediction difference. There is little we can do to aid conservation.
>
>
> > (3) Low-Data: whether a method works in the low-data regime.
> > - ABC: The decision process of ABC only relies on a few examples forming the minimal convex hull, hence the algorithm performs well under the low-data regime, making it generally applicable to many real-world data-scarce tasks.
> > - MFRL: In MFRL, the black-box value network and policy network can be designed to be sample-efficient.
> > - MBRL: In MBRL, sufficient data is always required to learn an accurate world model.
> > - BC: the performance of BC is highly dependent on the data quality. It is not designed for the low-data regime.
>
>
> > (4) Adaptive: whether the control behavior of a method can be adjusted according to additional constraints as clinical guidelines without modification or re-training.
> > - ABC: by changing reference examples, i.e., the decision corpus, during test time inference, ABC can seamlessly perform different types of decision-making according to user specifications.
> > - MFRL: In MFRL, when new data is used, a new value network and policy network need to be re-trained.
> > - MBRL: In MBRL, the world model construction is independent of the data, hence the decisions can be adaptive by changing a new planning algorithm on top of the world model. No model re-training is needed.
> > - BC: In BC, a new model needs to be trained with a specified type of decision corpus.
>
>
> > (5) Reward-Free: availability of extension to the strictly batched imitation setting where rewards are unavailable.
> > - ABC: ABC can be extended to the strictly batched imitation setting where rewards are unavailable.
> > - MFRL: In MFRL, the Q-values can not be calculated without the reward function.
> > - MBRL: In MBRL, the planning algorithms do not have a clear objective to optimize without reward signals.
> > - BC: BC is not affected by the absence of reward signals, because it does not need the reward to learn its policy.
>
> ---
> After enhancing the clarity in its motivation, problem definitions, and methodological flow, we sincerely hope that this updated version addresses the concerns on the presentation raised by reviewers fhj3 and fQBY.
>
> We would appreciate it if the reviewers kindly let us know if there were any further questions, including but not limited to the presentation. In the limited time remaining, we are still eager to do our utmost to address them!

---

### Decision · Program_Chairs · 2023-09-21

**Decision:**

Accept (poster)

**Comment:**

This paper is borderline, with an average score of 5.4 slightly below the acceptance threshold guidelines, and somewhat mixed reviews, with most reviewers recommending accept (7,6,5,5) and one (reviewer qevo) recommending borderline reject (4). One of reviewer qevo's main concerns is that the paper might be similar to offline RL but did not compare to offline RL baselines, a concern also identified by reviewer fQBY. The authors responded to this concern by elucidating several properties of their method not enjoyed by Offline RL, but also by adding some new results with Offline RL baselines for an experiment where it was applicable. This convinced reviewer fQBY to raise their score from a 3 to a 5, and reviewer qevo to raise their score from a 3 to a 4. Given that reviewer qevo is the only reviewer recommending reject, I attempted to engage them in further discussion during the discussion period but they did not respond. Reviewer fQBY did clarify why they raised their score, and mentioned that the additional experiments provided by the authors on multiple offline RL baselines were convincing. They also mentioned that they had initially misconstrued the optimization problem (Eq. 9), but found the further clarification provided by the authors to be helpful. In reading the detailed rebuttal provided by the authors, I found that they were able to point to additional experimental results in the appendix to address the other concerns raised by reviewer qevo. Given this, I am recommending an accept as a poster.